# FROM MOMENTS TO MODELS: GRAPHON MIXTURE-AWARE MIXUP AND CONTRASTIVE LEARNING

## ABSTRACT

Real-world graph datasets often consist of mixtures of populations, where graphs are generated from multiple distinct underlying distributions. However, modern representation learning approaches, such as graph contrastive learning (GCL) and augmentation methods like Mixup, typically overlook this mixture structure. In this work, we propose a unified framework that explicitly models data as a mixture of underlying probabilistic graph generative models represented by graphons. To characterize these graphons, we leverage graph moments (motif densities) to cluster graphs arising from the same model. This enables us to disentangle the mixture components and identify their distinct generative mechanisms. This model-aware partitioning benefits two key graph learning tasks: 1) It enables a graphon-mixture-aware mixup (GMAM), a data augmentation technique that interpolates in a semantically valid space guided by the estimated graphons, instead of assuming a single graphon per class. 2) For GCL, it enables model-adaptive and principled augmentations. Additionally, by introducing a new model-aware objective, our proposed approach (termed MGCL) improves negative sampling by restricting negatives to graphs from other models. We establish a key theoretical guarantee: a novel, tighter bound showing that graphs sampled from graphons with small cut distance will have similar motif densities with high probability. Extensive experiments on benchmark datasets demonstrate strong empirical performance. In unsupervised learning, MGCL achieves state-of-the-art results, obtaining the top average rank across eight datasets. In supervised learning, GMAM consistently outperforms existing strategies, achieving new state-of-the-art accuracy in 6 out of 7 datasets.

## 1 INTRODUCTION

Graphs provide a powerful framework for modeling complex relational data across diverse domains, including social networks (Yang et al., 2021; Kumar et al., 2022), bioinformatics (Balaji et al., 2022; Zhang et al., 2021; Azizpour et al., 2024), and wireless systems (Zhao et al., 2023; Olshevskyi et al., 2025). A central goal in graph machine learning is to learn representations that capture the underlying principles governing these networks for tasks like graph classification (Rey et al., 2025). Graphons, or graph limits, have emerged as a foundational tool for this, offering a continuous, non-parametric generative model that describes the large-scale structure of graphs (Lovász, 2012; Borgs et al., 2008). By representing the latent process from which graphs are sampled, graphons enable principled analysis and data augmentation (Navarro & Segarra, 2022; Han et al., 2022).

Given their utility, the accurate and efficient estimation of graphons (Chan & Airoldi, 2014; Xu et al., 2021; Xia et al., 2023) from observed network data is a central problem. This has spurred the development of several modern estimation techniques. For instance, methods like Scalable Implicit Graphon Learning (SIGL) leverage implicit neural representations and graph neural networks to learn a continuous graphon at arbitrary resolutions from large-scale graphs (Azizpour et al., 2025). Other recent work has shown that leveraging graph moments (subgraph counts) provides a computationally efficient path to scalable graphon estimation (Ramezanpour et al., 2025). However, a critical limitation of these approaches is that they are designed to estimate a single, unified graphon for the observed set of graphs. This assumption falters in real-world settings, where datasets frequently consist of a *mixture of populations*, with graphs generated from several distinct underlying

distributions, and as a result, modeling a single graphon may not capture highly heterogeneous graph data (Ramezanpour et al., 2025).

This limitation also extends to modern representation learning paradigms, which often overlook the shared or distinct underlying models of different graphs. For example, graph augmentation techniques such as G-mixup (Han et al., 2022) assume a single graphon per class to serve as the latent representation for mixing. However, graphs within the same class may arise from multiple generative processes, leading to heterogeneous latent spaces and semantically inconsistent augmentations. Similarly, in graph contrastive learning (You et al., 2020; 2021), each graph serves as an anchor, the positive sample whose representation is being learned, while all other graphs in the batch are treated as negatives. This neglects the possibility that some graphs originate from the same underlying generative model, making them false negatives and hindering representation quality.

In this work, we introduce a unified framework that explicitly addresses this challenge by modeling graph data as a mixture of underlying graphons. Our approach relies on the key insight that *densities of motifs* (empirical graph moments) serve as a powerful signature for the underlying generative model (Borgs et al., 2010). By leveraging these motif densities, we first partition the graph dataset into coherent clusters, where each cluster corresponds to a distinct component of the graphon mixture. This model-aware partitioning allows us to disentangle the generative mechanisms at play and forms the foundation for more principled machine learning techniques. Building on foundational results, we also establish a novel theoretical guarantee, showing that graphs sampled from graphons with a small cut distance will, with high probability, have similar motif densities.

Our main contributions are threefold:

- We introduce a moment-based clustering method to partition heterogeneous graph datasets into clusters, each corresponding to a distinct underlying graphon, providing an interpretable estimation of graphon mixtures.

- For supervised settings, we propose a graphon mixture-aware mixup (GMAM) that identifies multiple underlying generative models within each class of data and performs mixing based on these models, ensuring semantically consistent augmented graphs.

- For unsupervised learning, we develop a moment-aware contrastive framework (MGCL) that leverages graphon-specific augmentations and model-aware negative sampling to improve augmentations and reduce false negatives.

## 2 PRELIMINARY

### 2.1 GRAPHON

A graphon is defined as a bounded, symmetric, and measurable function $W : [0,1]^2 \rightarrow [0,1]$ (Lovász, 2012; Avella-Medina et al., 2018). By construction, a graphon acts as a *generative model for random graphs*, allowing the sampling of graphs that exhibit similar structural properties. To generate an undirected graph $\mathcal{G}$ with $N$ nodes from a given graphon $W$, the process consists of two main steps: (1) assigning each node a latent variable drawn uniformly at random from the interval $[0,1]$, and (2) connecting each pair of nodes with a probability given by evaluating $W$ at their respective latent variable values. Formally, the steps are as follows:

$$\begin{aligned} \boldsymbol{\eta}(i) &\sim \text{Uniform}([0,1]), \quad \forall\, i = 1, \cdots, N, \\ \mathbf{A}(i,j) &\sim \text{Bernoulli}\left(W(\boldsymbol{\eta}(i), \boldsymbol{\eta}(j))\right), \quad \forall\, i,j = 1, \cdots, N, \end{aligned} \tag{1}$$

where the latent variables $\boldsymbol{\eta}(i) \in [0,1]$ are independently drawn for each node $i$. The dissimilarity between two graphons, $W_1$ and $W_2$, is measured by the cut distance, denoted as $d_{\text{cut}}(W_1, W_2)$ (Lovász, 2012).

**Graphon estimation.** The generative process in equation 1 can also be viewed in reverse: given a collection of graphs (represented by their adjacency matrix) $\mathcal{D} = \{\mathbf{A}_t\}_{t=1}^{M}$ that are sampled from an *unknown* graphon $W$, estimate $W$. Several methods have been proposed for this task (Chan & Airoldi, 2014; Airoldi et al., 2013; Xu et al., 2021; Xia et al., 2023; Azizpour et al., 2025; Ramezanpour et al., 2025). We focus on SIGL (Azizpour et al., 2025), a resolution-free method that, in addition to estimating the graphon, also *infers the latent variables* $\boldsymbol{\eta}$, making it particularly useful for

model-driven augmentation. This method parameterizes the graphon using an implicit neural representation (INR) (Sitzmann et al., 2020), a neural architecture defined as $f_\phi(x, y) : [0, 1]^2 \to [0, 1]$ where the inputs are coordinates from $[0, 1]^2$ and the output approximates the graphon value $W$ at a particular position. More details of SIGL are provided in Appendix D.

**Motif densities from graphons**    A graphon $W$ provides the theoretical expectation for the density of any subgraph, also known as a motif $F$. The non-induced homomorphism density, $t(F, W)$, counts all occurrences of a motif where at least the specified edges are present, regardless of any additional edges that might exist between the motif's vertices. The non-induced density is given as follows

$$t(F, W) = \int_{[0,1]^k} \prod_{(i,j) \in \mathcal{E}_F} W(\eta_i, \eta_j) \prod_{l \in \mathcal{V}_F} d\eta_l. \tag{2}$$

The power of this framework lies in its connection to observable data; the theoretical value of $t(F, W)$ can be accurately estimated by the empirical motif counts found in a large graph sampled from the graphon $W$ (Lovász, 2012). This makes motif densities a powerful tool for linking continuous graphon models to discrete, real-world networks.

## 2.2 GRAPH MIXUP

Mixup (Zhang et al., 2018; Verma et al., 2019) has been successfully adapted to graphs in several works (Han et al., 2022; Navarro & Segarra, 2023). The key idea is to augment the dataset by interpolating existing samples. In its standard form, mixup generates a new sample by linearly combining two randomly chosen inputs and their labels:

$$x_{\text{new}} = \lambda x_i + (1 - \lambda)x_j, \quad y_{\text{new}} = \lambda y_i + (1 - \lambda)y_j, \tag{3}$$

where $(x_i, y_i)$ and $(x_j, y_j)$ are two samples with one-hot labels. In the graph domain, G-Mixup extends this idea by operating at the level of graphons (Han et al., 2022; Navarro & Segarra, 2023). Given two classes, it first estimates their graphons, then interpolates between them, and finally generates new graphs and labels from the mixed graphon:

$$\text{Graphon estimation: } \mathcal{D}_0 \to W_\mathcal{G}, \quad \mathcal{D}_1 \to W_\mathcal{H}, \tag{4}$$

$$\text{Graphon mixup: } W_\mathcal{I} = \lambda W_\mathcal{G} + (1 - \lambda)W_\mathcal{H}, \tag{5}$$

$$\text{Graph generation: } \{I_1, \ldots, I_m\} \overset{\text{i.i.d.}}{\sim} \mathbb{G}(K, W_\mathcal{I}), \tag{6}$$

$$\text{Label mixup: } \mathbf{y}_\mathcal{I} = \lambda \mathbf{y}_\mathcal{G} + (1 - \lambda)\mathbf{y}_\mathcal{H}. \tag{7}$$

## 2.3 GRAPH CONTRASTIVE LEARNING

GCL (Veličković et al., 2019; You et al., 2020; 2021; Suresh et al., 2021) aims to learn discriminative node or graph-level embeddings in a self-supervised manner, without relying on explicit labels. Given a collection of graphs $\{\mathcal{G}_t\}_{t=1}^T$, the goal of graph-level GCL is to train an encoder $\mathcal{E}_\theta(\cdot)$ so that it produces expressive representations or embeddings. The encoder outputs an embedding per graph $\mathcal{G}_t$, denoted by $\mathbf{z}_t = \mathcal{E}_\theta(\mathbf{A}_t, \mathbf{X}_t)$, where $\mathbf{z}_t \in \mathbb{R}^{1 \times F}$. InfoNCE-based methods are widely adopted in this setting You et al. (2020); Suresh et al. (2021); You et al. (2021). These methods generate an augmented view of the input graph through transformations such as edge perturbation, feature masking, or subgraph sampling. Following this, let $\mathbf{z}_t$ and $\tilde{\mathbf{z}}_t$ denote the graph-level representations of the original and augmented views, respectively. The InfoNCE loss is then used to bring $\mathbf{z}_t$ and $\tilde{\mathbf{z}}_t$ (positive pair) closer in the embedding space while pushing them apart from representations of all other graphs in the batch ($\tilde{\mathbf{z}}_{t'}$, for all $t' \neq t$, i.e., negative samples). Formally, the parameters of the encoder are achieved by minimizing the following loss function

$$\ell_t = -\log \frac{\exp(\text{sim}(\mathbf{z}_t, \tilde{\mathbf{z}}_t)/\tau)}{\sum_{t'=1, t' \neq t}^L \exp(\text{sim}(\mathbf{z}_t, \tilde{\mathbf{z}}_{t'})/\tau)}, \quad \mathcal{L}_{\text{all}} = \frac{1}{L} \sum_{t=1}^L \ell_t. \tag{8}$$

## 3 METHODS

We first present our unified framework for estimating the multiple underlying data distributions of graphs as a graphon mixture in Section 3.1. We then introduce novel methods that adapt and leverage

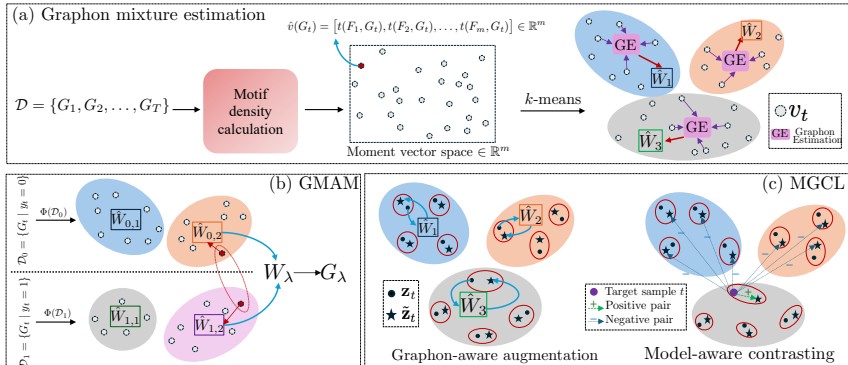

Figure 1: Overview of the proposed framework. (a) Graphon mixture estimation via motif moment vectors, (b) Graphon mixture–aware mixup for data augmentation, (c) Model-aware GCL leveraging graphon-informed augmentations and model-aware contrastive loss.

these estimated graphons in the context of graph mixup and graph contrastive learning, as detailed in Sections 3.2 and 3.3, respectively. An overview of these steps is shown in Figure 1, and a detailed algorithm of these three steps is included in Appendix A.

## 3.1 GRAPHON MIXTURE ESTIMATION

The goal of this step, which is the core of our methodology, is to recover the multiple underlying generative models of the data, represented as a set of graphons $\{W_i\}_{i=1}^{K}$. We assume that each observed graph $G_t$ in the dataset is sampled from one of these graphons, though the assignment is unknown. Hence, the task reduces to identifying groups of graphs that are likely to originate from the same underlying distribution, and subsequently estimating a representative graphon for each group.

To formalize this idea, we leverage the concept of *moment vectors*. For a graphon $W$, its motif densities provide expectations of subgraph counts (cf. equation 2). Let $\mathcal{F} = \{F_1, \ldots, F_m\}$ denote a fixed family of motifs. The *moment vector* of a graphon $W$ with respect to $\mathcal{F}$ is defined as

$$v(W) = \big[t(F_1, W), t(F_2, W), \ldots, t(F_m, W)\big] \in \mathbb{R}^m. \tag{9}$$

Similarly, for an observed graph $G$ sampled from $W$, we compute its empirical moment vector $\hat{v}(G)$ by normalizing motif counts in $G$. We present next a result to bound the difference between the empirical moment vectors of two graphs.

**Theorem 1** (Bounding Empirical Density Difference by Graphon Distance). *Let $G_1$ and $G_2$ be $n$-vertex graphs sampled from graphons $W_1$ and $W_2$, respectively, where $d_{cut}(W_1, W_2) \leq \epsilon$. Let $\mathcal{F}$ be a fixed family of motifs of bounded size. For any motif $F \in \mathcal{F}$ with $k = v(F)$ vertices and $e(F)$ edges, and for any target failure probability $\eta > 0$, we have*

$$|t(F, G_1) - t(F, G_2)| \leq \underbrace{e(F)\epsilon}_{\text{Graphon Difference}} + 2\underbrace{\left(\sqrt{\frac{1}{2m}\log\frac{4}{\eta}} + \frac{e(F)}{\sqrt{n(n-1)}}\sqrt{2\log\frac{4}{\eta}}\right)}_{\text{Sampling Error}}, \tag{10}$$

*with probability at least $1 - 2\eta$, where $m = \left\lfloor \frac{n}{k} \right\rfloor$. The proof is provided in Appendix B.*

The bound in Equation equation 10 on the total difference in empirical motif densities is comprised of two components. The first term, the **Graphon Difference**, arises from the intrinsic structural dissimilarity between the underlying graphons. The second term, the **Sampling Error**, represents a novel, tighter concentration bound that quantifies the statistical uncertainty from the random graph generation process. This novel bound offers a significant improvement over classical approaches that can be derived from the literature (Lovász, 2012; Borgs et al., 2008), as we demonstrate in Appendix B, where a full derivation and comparison are provided.

Theorem 1 reveals that if the underlying graphons $W_1$ and $W_2$ are close in cut distance, i.e., small $\epsilon$, then their corresponding empirical moment vectors, $\hat{v}(G_1)$ and $\hat{v}(G_2)$, are also close with high

probability. Alternatively, it also follows that if the moment vectors of two graphs $G_1$ and $G_2$ are observed to be significantly different, we can conclude with high confidence that they were generated from structurally different underlying graphons. Formally, if $|t(F, G_1) - t(F, G_2)|$ is large for some motif $F$, it implies that $d_{\mathrm{cut}}(W_1, W_2)$ must also be large. By applying a union bound across all motifs in the family $\mathcal{F}$, this principle extends to the full moment vector: a large Euclidean distance $\|\hat{v}(G_1) - \hat{v}(G_2)\|_2$ implies that $G_1$ and $G_2$ likely originate from different graphon distributions. This justifies the use of motif-based embeddings for graph clustering, as distinct generating processes produce measurably distinct motif fingerprints. Further theoretical analyses are included in Appendix B.

Based on the above theorem, we assign each graph $G_t$ a feature vector $v_t = \hat{v}(G_t) \in \mathbb{R}^m$, and collect them as $\{v_t\}_{t=1}^T$. We then apply $k$-means clustering to these vectors in order to group graphs according to their latent generative models. Let $\{C_1, \ldots, C_K\}$ denote the resulting clusters. Within each cluster $C_k$, the graphs share similar moment vectors, suggesting they arise from a common graphon $W_k$. Then, we aim to estimate a graphon for each cluster of graphs. To mitigate boundary effects (i.e., graphs near the edges of clusters that may not be well represented), we select the $L$ graphs closest to the cluster centroid in moment space. For each cluster $C_k$, we estimate the underlying graphon $W_k$ using these $L$ graphs. Any graphon estimation algorithm can be applied; in our implementation, we adopt SIGL (Azizpour et al., 2025), which additionally provides the latent node variables $\boldsymbol{\eta}$ alongside the graphon estimate. Finally, each graph $G_t$ in cluster $C_k$ is associated with the estimated graphon $W_k$. This provides us with a graphon mixture model $\{W_1, \ldots, W_K\}$, where each observed graph is assigned to exactly one mixture component. To align with the dataset size, we set $K = \log T$.

In summary, we can view the overall pipeline as a mapping from a collection of graphs to a set of estimated graphons together with an assignment function. Let $\mathcal{D} = \{G_1, G_2, \ldots, G_T\}$ denote the dataset of observed graphs. Our procedure outputs (i) a set of estimated graphons $\{\hat{W}_1, \ldots, \hat{W}_K\}$ and (ii) an assignment function

$$\pi : \mathcal{D} \to \{1, \ldots, K\},$$

such that each graph $G_t$ is mapped to one of the estimated graphons $\hat{W}_{\pi(G_t)}$. Equivalently, the pipeline defines a function

$$\Phi : \mathcal{D} \longrightarrow \big(\{\hat{W}_1, \ldots, \hat{W}_K\}, \pi\big), \tag{11}$$

where $\Phi$ encapsulates the steps of (i) computing empirical moment vectors, (ii) clustering in moment space, and (iii) estimating graphons for each cluster.

## 3.2 GRAPHON MIXTURE-AWARE MIXUP (GMAM)

In this section, we describe how to leverage the graphon mixture for graph data augmentation under the mixup framework. In contrast to existing G-mixup, which assumes a single graphon per class, our approach disentangles the multiple generative models that may exist within each class. This enables a more fine-grained and semantically valid interpolation strategy.

Consider a dataset of graphs with $C$ classes, $\mathcal{D} = \{(G_t, y_t)\}_{t=1}^T$, where $y_t \in \{1, \ldots, C\}$. For each class $i$, we first collect its subset of graphs

$$\mathcal{D}_i = \{G_t \mid y_t = i\}. \tag{12}$$

We then apply the operator $\Phi$ from Section 3.1 to estimate the graphon mixture within each class. Formally, this yields

$$\Phi(\mathcal{D}_i) \longrightarrow \big(\{\hat{W}_{i,1}, \ldots, \hat{W}_{i,K_i}\}, \pi_i\big), \tag{13}$$

where $\{\hat{W}_{i,1}, \ldots, \hat{W}_{i,K_i}\}$ denotes the mixture of graphons estimated for class $i$, and $\pi_i$ is the assignment of each graph in $\mathcal{D}_i$ to its underlying graphon.

To perform GMAM, we first sample two graphs $G_a \in \mathcal{D}_i$ and $G_b \in \mathcal{D}_j$ from classes $i \neq j$. Instead of directly interpolating their raw structure, we trace back to their associated graphons, $\hat{W}_{i,\pi_i(G_a)}$ and $\hat{W}_{j,\pi_j(G_b)}$. We then construct a mixed graphon

$$W_\lambda = \lambda \hat{W}_{i,\pi_i(G_a)} + (1 - \lambda)\hat{W}_{j,\pi_j(G_b)}. \tag{14}$$

Finally, we generate a new graph $G_\lambda \sim \mathbb{G}(n, W_\lambda)$ according to the stochastic sampling process in equation 1, where $n$ is chosen to match the scale of the dataset. The associated label is interpolated in the standard mixup fashion: $\mathbf{y}_\lambda = \lambda \mathbf{y}_i + (1 - \lambda)\mathbf{y}_j$, where $\mathbf{y}_i, \mathbf{y}_j$ are one-hot encoded labels.

By disentangling class distributions into mixtures of graphons, we preserve finer semantic structures during augmentation and avoid the unrealistic assumption that all graphs within a class share a single generative model.

### 3.3 Model-aware Graph Contrastive Learning (MGCL)

We now extend the graphon mixture framework to contrastive learning. Given an unlabeled dataset of graphs $\mathcal{D} = \{G_t\}_{t=1}^T$, we first apply the operator $\Phi$ (Section 3.1) to obtain the graphon mixture $\{\hat{W}_1, \ldots, \hat{W}_K\}$ along with the assignment $\pi$ that maps each graph $G_t$ to its generating graphon $\hat{W}_{\pi(G_t)}$. This mapping benefits contrastive learning in two key ways: (1) it enables a principled graphon-aware augmentation strategy that generates meaningful positive pairs, and (2) it supports a model-aware loss function.

**Graphon-aware augmentation.**  For each cluster $k$, the estimated graphon $\hat{W}_k$ characterizes the generative distribution of graphs in that cluster.  We use this distribution to design a graphon-informed augmentation procedure. Given a graph $G_t$ with adjacency matrix $\mathbf{A}_t$ that belongs to cluster $C_t$, we randomly select a subset $E_{\text{sel}}$ of $r\%$ of all node pairs. For each selected pair $(i, j) \in E_{\text{sel}}$, the corresponding adjacency entry is resampled according to the probability given by the cluster's graphon:

$$\tilde{\mathbf{A}}_t(i, j) \sim \text{Bernoulli}\big(\hat{W}_{C_t}(\eta_i, \eta_j)\big), \quad \tilde{\mathbf{A}}_t(j, i) = \tilde{\mathbf{A}}_t(i, j), \tag{15}$$

where $\eta_i, \eta_j \in [0, 1]$ denote the latent positions associated with nodes $i$ and $j$.  For all pairs $(i, j) \notin E_{\text{sel}}$, we retain the original edges: $\tilde{\mathbf{A}}_t(i, j) = \mathbf{A}_t(i, j)$.  Note that, to leverage graphons for augmentation and edge resampling, we require the latent variables of the nodes ($\eta$) in order to query the edge probabilities from the graphon.  Since SIGL is specifically designed to estimate graphons together with the latent node positions, we adopt SIGL (Azizpour et al., 2025) for graphon estimation.  Nevertheless, other graphon estimation methods (Chatterjee, 2015; Chan & Airoldi, 2014) can also be employed, as they implicitly assign latent variables by sorting nodes according to degree and using the degree as a proxy for the latent position.

This process injects structure-aware perturbations guided by the estimated generative model.  Unlike naive random perturbations, graphon-aware augmentation assigns edge-specific probabilities informed by the graphon, producing more faithful augmented views.  Passing the original graph and its graphon-aware augmentation through the encoder $\mathcal{E}_\theta$ with node features $\mathbf{X}_t$ yields the corresponding representations $\mathbf{z}_t = \mathcal{E}_\theta(\mathbf{A}_t, \mathbf{X}_t)$ and $\tilde{\mathbf{z}}_t = \mathcal{E}_\theta(\tilde{\mathbf{A}}_t, \mathbf{X}_t)$; see Section 2.3.

**Model-aware contrasting.**  We modify the InfoNCE objective (Oord et al., 2018) to account for graphon mixture assignments.  Given a batch of $L$ graphs, the loss for anchor $t$ and the overall loss function is

$$\ell_t = -\log \frac{\exp(\text{sim}(\mathbf{z}_t, \tilde{\mathbf{z}}_t)/\tau)}{\sum\limits_{t'=1, C_{t'} \neq C_t}^{L} \exp(\text{sim}(\mathbf{z}_t, \tilde{\mathbf{z}}_{t'})/\tau)}, \quad \mathcal{L}_{\text{all}} = \frac{1}{L}\sum_{t=1}^{L} \ell_t, \tag{16}$$

where $\text{sim}(\cdot, \cdot)$ is a similarity measure (e.g., cosine), $\tau$ is the temperature, and $C_t$ is the cluster index of $G_t$.  Unlike standard InfoNCE, which pushes a graph away from all other graphs (including those with the same underlying model), our formulation only contrasts against graphs from *different* graphons. The following result clarifies how our loss interpolates between model-level and instance-level contrasting:

**Theorem 2** (Lower bound of model-aware loss). *For every graph $t$ let $m_t := |\{k : C_k \neq C_t\}|$, and $\bar{\mathbf{z}}^{(\neg C_t)} := \frac{1}{m_t}\sum_{C_k \neq C_t} \tilde{\mathbf{z}}_k$ be the centroid of negative samples from clusters other than $C_t$. Then,*

$$\ln m_t + \text{sim}(\mathbf{z}_t, \bar{\mathbf{z}}^{(\neg C_t)}) - \text{sim}(\mathbf{z}_t, \tilde{\mathbf{z}}_t) \;\leq\; \ell_t. \tag{17}$$

The proof is provided in Appendix C. This lower bound shows that by minimizing our loss function, each graph representation $\mathbf{z}_t$ is contrasted against the centroid of unrelated models (smaller

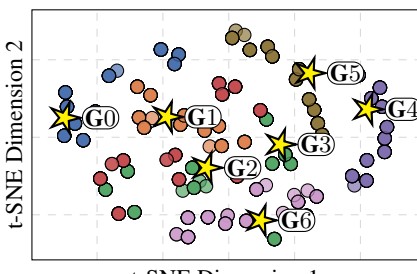 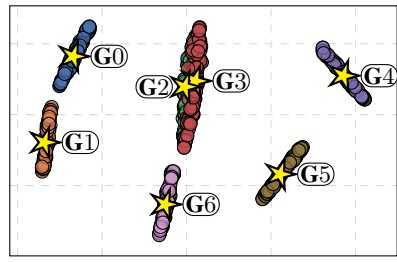

Figure 2: t-SNE embedding of graphs. Left: *Varying* size, $n \sim U[75, 300]$. Right: *Fixed* size, $n = 200$. Each color represents different graphon.

values for $\text{sim}(\mathbf{z}_t, \bar{\mathbf{z}}^{(\neg C_t)})$, rather than against the centroid of the entire dataset. Moreover, positive samples $\tilde{\mathbf{z}}_t$ are improved by using graphon-aware augmentations, which preserve the semantics of the underlying generative process, while negatives are improved by excluding graphs from the same mixture, reducing false negatives. As a result, the encoder learns to align each graph representation with its generating model while scattering it away from other models. Thus, our method can be interpreted as contrasting each graph from the *negative models* rather than individual graphs and pushing each graph toward its *positive model*, yielding representations that are both more discriminative and semantically faithful.

## 4 EXPERIMENTS

We evaluate our proposed pipeline along three main directions. First, in Section 4.1, we test whether clustering based on moment vectors can successfully separate graphs generated from different underlying models. Next, in Sections 4.2 and 4.3, we assess the effectiveness of our mixture-aware extensions: GMAM and MGCL, respectively. A description of the datasets, baselines, experimental setups, and hyperparameters for all three types of experiments is provided in Appendix E.

### 4.1 SYNTHETIC EXPERIMENT

To empirically validate the theoretical framework presented in Section 3.1, we conduct a synthetic experiment designed to test the efficacy of moment vectors in distinguishing and clustering graphs sampled from a mixture of different graphon distributions. The core hypothesis is that graphs generated from distinct graphons will exhibit measurably different motif densities, forming separable clusters in the moment vector space.

**Experimental setup.** We generate a dataset composed of graphs sampled from a mixture of $K = 7$ distinct, predefined graphon models, $\{W_k\}_{k=0}^{6}$. These models, include a range of functional forms to ensure structural diversity, generating graphs of varying density through models that are, for example, linear or exponential in nature. For each graphon model $W_k$, we sample a collection of graphs, creating a dataset where the ground-truth generating model for each graph is known.

To analyze the impact of graph size on the concentration of empirical moments around their theoretical means (cf. Theorem 1), we structure our experiment into two distinct scenarios: (i) *Varying* size, where $n$ is drawn uniformly from $[75, 300]$ (as in SIGL (Azizpour et al., 2025) and MomentNet (Ramezanpour et al., 2025)); and (ii) *Fixed* size, with $n = 200$. For each scenario, we generate a balanced dataset with an equal number of graphs from each of the seven graphon classes. For each sampled graph, we compute its empirical moment vector using the densities

| Method | Varying | Fixed |
|---|---|---|
| Theory | 81.4 | 82.9 |
| GCN | 58.6 | 64.7 |
| GIN | 25.7 | 60.9 |
| Graph2Vec | 28.6 | 62.0 |
| DeepWalk | 25.7 | 28.9 |
| Spectral | 22.9 | 21.7 |
| MBC (our) | **80.0** | **79.3** |

Table 1: Clustering accuracy (in %) with different embeddings.

of all connected motifs with up to 4 nodes, resulting in feature vectors of size 9. We then apply the clustering algorithm introduced in Section 3.1 to cluster these vectors and measure the accuracy by comparing the assignments to the ground-truth labels. This directly tests the ability of moment vectors to partition a mixture of graphs generated from different underlying models.

Table 2: Classification accuracy (%) of different mixup methods compared with GMAM across multiple datasets. The top performing method is **boldfaced**, and the second best is underlined. *: Result not available.

| Dataset | IMDB-B | IMDB-M | REDD-B | REDD-M5 | REDD-M12 | COLLAB | AIDS |
|---|---|---|---|---|---|---|---|
| Vanilla | 71.30±4.36 | 48.80±2.54 | 89.15±2.47 | 53.17±2.26 | 49.95±0.98 | 79.39±1.24 | 98.00±1.20 |
| DropEdge | 70.50±3.80 | 48.73±4.08 | 87.45±3.91 | 54.11±1.94 | 49.77±0.76 | 78.32±1.31 | 92.87±1.45 |
| DropNode | 72.00±6.97 | 45.67±2.59 | 88.60±2.52 | 53.97±2.11 | 49.95±1.70 | 80.01±1.66 | 96.25±1.24 |
| SubMix | 71.70±6.20 | 49.80±4.01 | 90.45±1.93 | 54.27±2.92 | 42.58±12.30 | 80.11±0.75 | 96.18±1.33 |
| M-Mixup | 72.00±5.14 | 48.67±5.32 | 87.70±2.50 | 52.85±1.03 | 49.81±0.80 | 77.00±2.20 | * |
| S-Mixup | 73.40±6.26 | 50.13±4.34 | 90.55±2.11 | 55.19±1.99 | 49.51±1.59 | * | * |
| *G*-Mixup | 72.40±5.64 | 49.93±2.82 | 90.20±2.84 | 54.33±1.99 | 49.72±0.48 | 79.05±1.25 | 97.8±0.90 |
| SIGL | 73.95±2.64 | 50.70±1.41 | 91.93±0.94 | 55.82±1.35 | 49.73±0.62 | 80.15±0.60 | 97.93±0.96 |
| MomentMixup | 74.30±2.70 | 50.95±1.93 | 91.80±1.20 | 56.09±1.62 | 49.83±1.01 | 79.75± 0.59 | **98.50**±0.60 |
| GMAM | **74.45**±1.15 | **51.03**±1.63 | **92.25**±0.82 | **56.46**±0.95 | **50.18**±0.50 | **80.25**±0.52 | 98.20±0.51 |

**Results.** To visualize the structure of the moment vector space, we use t-SNE to project the 9-dimensional moment vectors into a 2D plane, as shown in Figure 2. In these plots, each point represents a graph sampled from one of seven graphon distributions, and the stars denote the ground-truth moment vectors $v(W_k)$ for each underlying graphon $W_k$. The visualizations empirically confirm our theoretical framework. According to Theorem 1, the distance between the empirical moment vectors of two sampled graphs, $\hat{v}(G_1)$ and $\hat{v}(G_2)$, is influenced by two primary factors, namely the distance between their underlying graphons, $d_{\text{cut}}(W_1, W_2)$, and a sampling error term which decreases as the number of nodes $n$ grows. In the *varying* size scenario (Figure 2(a)), the higher variance in graph sizes leads to more diffuse clusters, as smaller graphs introduce larger sampling errors. Conversely, in the *fixed* size scenario with $n = 200$ (Figure 2(b)), the empirical vectors concentrate more tightly around their respective ground-truth means, forming more distinct and separable clusters. The separation between clusters is determined by the distance between their underlying graphons. Table 6 (Appendix E) reports the pairwise Gromov–Wasserstein (GW) distances, used here as a practical proxy for the cut distance since the latter can be relaxed to the GW distance of step functions (Xu et al., 2021). As expected, graphons 2 and 3 have a very small GW distance of $0.024$, reflected in the overlap of their green and red clusters, while graphons with larger distances, such as 0 and 4 ($0.530$), yield well-separated clusters.

Quantitatively, our clustering performance is reported in Table 1. The Theory-Based accuracy is computed by assigning each graph to the cluster of the nearest ground-truth moment vector in Euclidean space, providing an upper bound on performance. Other methods apply k-means clustering on the embeddings obtained by different methods. Our proposed method, MBC (moment-based clustering), achieves accuracies of $0.800$ and $0.793$ for the varying and fixed size settings, respectively, outperforming all other embedding methods and closely approaching the theory-based scores of $0.814$ and $0.829$. This demonstrates that moment vectors provide a robust foundation for clustering graphs generated from different underlying distributions. The higher accuracy in the fixed-size setting directly reflects the tighter cluster formations observed in the t-SNE visualization, further validating the concentration properties outlined in our theory. Further details of different embedding methods are provided in Appendix E.5.

## 4.2 GMAM

In this section, we evaluate the effectiveness of our proposed mixture-aware graph mixup method in Section 3.2 on seven real-world datasets from the TUDatasets benchmark (Morris et al., 2020). These include two bioinformatics datasets (PROTEINS, AIDS), one molecular dataset (NCI1), and four social network datasets (IMDB-BINARY, IMDB-MULTI, REDDIT-BINARY, REDDIT-MULTI-5K). We compare against a broad range of augmentations and state-of-the-art mixup variants. Full details on baselines are provided in Appendix E.

For fairness, we adopt the same GNN structure (depth and architecture) across all methods, along with identical training hyperparameters. All steps of graphon mixture estimation and the subsequent mixup augmentation are performed strictly within the training set The augmentation ratio (i.e., the proportion of generated graphs added to training) is also fixed consistently across methods. Table 2 reports the results. Our method consistently improves performance across multiple datasets, achieving the highest accuracy on five of eight benchmarks, ranking second on two others, and third on the remaining one. These results demonstrate that incorporating mixture-aware graphon models leads to more effective augmentations by disentangling multiple generative patterns within each class.

Table 3: Unsupervised representation learning classification accuracy (%) on TU datasets. A.R denotes the average rank of the results. The top performing method is **boldfaced**, and the second best is underlined.

| Methods | NCI1 | PROTEINS | DD | MUTAG | COLLAB | RDT-B | RDT-M5K | IMDB-B | A.R. |
|---|---|---|---|---|---|---|---|---|---|
| InfoGraph | 76.20 ±1.0 | 74.44 ±0.3 | 72.85 ±1.8 | 89.01 ±1.1 | 70.65 ±1.1 | 82.50 ±1.4 | 53.46 ±1.0 | **73.03** ±0.9 | 7.37 |
| GraphCL | 77.87 ±0.4 | 74.39 ±0.4 | 78.62 ±0.4 | 86.80 ±1.3 | 71.36 ±1.1 | 89.53 ±0.8 | 55.99 ±0.3 | 71.14 ±0.4 | 6.25 |
| MVGRL | 76.64 ±0.3 | 74.02 ±0.3 | 75.20 ±0.4 | 75.40 ±7.8 | 73.10 ±0.6 | 82.00 ±1.10 | 51.87 ±0.6 | 63.60 ±4.2 | 9.12 |
| JOAO | 78.07 ±0.5 | 74.55 ±0.4 | 77.32 ±0.5 | 87.35 ±1.0 | 69.50 ±0.3 | 85.29 ±1.3 | 55.74 ±0.6 | 70.21 ±3.1 | 7.75 |
| JOAOv2 | 78.36 ±0.5 | 74.07 ±1.1 | 77.40 ±1.1 | 87.67 ±0.8 | 69.33 ±0.3 | 86.42 ±1.4 | 56.03 ±0.3 | 70.83 ±0.3 | 6.87 |
| AD-GCL | 73.90 ±0.8 | 73.30 ±0.5 | 75.80 ±0.9 | 88.70 ±1.9 | 72.00 ±0.6 | 90.10 ±0.9 | 54.30 ±0.3 | 70.20 ±0.7 | 7.50 |
| AutoGCL | 78.32 ±0.5 | 69.73 ±0.4 | 75.75 ±0.6 | 85.15 ±1.1 | 71.40 ±0.7 | 86.60 ±1.5 | 55.71 ±0.2 | 72.00 ±0.4 | 7.25 |
| simGRACE | 79.10 ±0.4 | 75.30 ±0.1 | 77.40 ±1.1 | 89.00 ±1.3 | 71.70 ±0.8 | 89.50 ±0.9 | 55.90 ±0.3 | 71.30 ±0.8 | 4.37 |
| RGCL | 78.10 ±1.0 | 75.00 ±0.4 | 78.90 ±0.5 | 87.70 ±1.0 | 71.00 ±0.7 | 90.30 ±0.6 | **56.40** ±0.4 | 71.90 ±0.9 | 4.25 |
| DRGCL | 78.70 ±0.4 | 75.20 ±0.6 | 78.40 ±0.7 | 89.50 ±0.6 | 70.60 ±0.8 | **90.80** ±0.3 | 56.30 ±0.2 | 72.00 ±0.5 | 3.37 |
| MGCL | **79.18** ±0.48 | **75.54** ±0.31 | **79.07** ±0.24 | **90.03** ±1.32 | **72.22** ±0.72 | 90.46 ±0.14 | 56.08 ±0.24 | 72.28 ±0.52 | **1.62** |

Thus, our method provides more semantically faithful interpolations, which translate into stronger downstream performance.

### 4.3 MGCL

**Setup.** We compare MGCL with a variety of state-of-the-art GCL baselines. The full set of competing methods is detailed in Appendix E. We evaluate on eight datasets from the TUDataset (Morris et al., 2020) collection. The experiments follow the linear evaluation protocol (Peng et al., 2020), where models are first trained in an unsupervised manner, and the resulting embeddings are subsequently used for downstream tasks. We adopt the evaluation protocol from (Sun et al., 2020), performing 10-fold cross-validation on each dataset. The resulting graph-level embeddings are used to train an SVM classifier, and we report the average performance across the folds. To summarize performance, we assign a rank to each method based on its performance on each dataset, and the average rank (A.R.) is then computed as the mean of these ranks across all datasets.

**Results.** As shown in Table 3, MGCL achieves strong performance across a wide range of graph-level benchmarks from the TUDataset collection (Morris et al., 2020). MGCL ranks first on five out of eight datasets and second on two of the other datasets. For the IMDB-B dataset, Info-Graph performs best, suggesting that augmentation-based strategies may be less effective for this particular benchmark. Nonetheless, MGCL outperforms other augmentation-based methods, such as GraphCL and JOAO, and more recent, powerful methods such as DRGCL and simGRACE. Most notably, MGCL obtains the lowest average rank (A.R.) of 1.62, outperforming all competing baselines. These results highlight MGCL's ability to learn generalizable and transferable graph-level representations. Further experiments on the effectiveness of our model-aware loss function are included in Appendix F, where we observe that clustering indeed reduces the false negative rate.

## 5 CONCLUSION

We proposed a unified framework for inferring mixtures of underlying generative models (graphons) from observed graphs. Instead of assuming a single distribution per dataset, our approach disentangles multiple generative models and leverages this structure for downstream learning. We developed three components: (i) graphon mixture estimation via motif moment vectors clustering, theoretically supported by concentration guarantees for moments, (ii) GMAM, a graphon mixture-aware mixup for model-level data augmentation, and (iii) MGCL, a contrastive learning framework combining graphon-informed augmentations with a model-aware loss. Empirically, we showed that moment-based clustering recovers ground-truth models, GMAM achieves state-of-the-art results on seven benchmarks, and MGCL reduces false negatives and yields more semantically faithful representations. While our framework is effective, its performance may be influenced by factors such as graph size. For smaller graphs in particular, higher sampling variance in motif counts can cause their embeddings to deviate from their theoretical means, making it challenging to reliably identify their true underlying generative distribution. Future work could extend this framework to use more expressive generative models (e.g., higher-order kernels). Additionally, the strong separability of motif embeddings suggests a promising avenue for graph-level anomaly detection, where graphs that are outliers in the moment space could be identified as originating from anomalous generative processes.

## REPRODUCIBILITY STATEMENT

The model architectures, training hyperparameters, and experimental settings are detailed in Section 4, with additional implementation information provided in Appendix E. Proofs of the theoretical results are given in Sections 3.1 and 3.3, with all assumptions stated explicitly in Appendices B.1 and C. Dataset statistics are also reported in Appendix E. Finally, the source code and scripts are included in the supplementary materials, along with instructions to reproduce all experiments and results.

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

## A    ALGORITHMS

We present the three core algorithms that constitute our framework. The first, Algorithm 1, details the operator $\Phi$ for estimating a graphon mixture model from a graph dataset by clustering graphs based on their motif densities. The subsequent algorithms leverage this estimated mixture for downstream tasks: Algorithm 2 introduces a novel mixup-based data augmentation strategy (GMAM) that interpolates between the generative graphon models, while Algorithm 3 presents a model-aware contrastive learning method (MGCL) that uses the mixture for both principled data augmentation and a more effective negative sampling strategy.

---

**Algorithm 1:** Graphon Mixture Estimation (Operator $\Phi$)

**Input** : Graph dataset $\mathcal{D} = \{G_t\}_{t=1}^{T}$; motif family $\mathcal{F} = \{F_1, \ldots, F_m\}$; #clusters $K = \log T$; per-cluster refinement size $L$; graphon estimator (e.g., SIGL).

**Output:** Estimated graphons $\{\hat{W}_1, \ldots, \hat{W}_K\}$; assignment $\pi : \mathcal{D} \to \{1, \ldots, K\}$.

1 **for** $t = 1$ **to** $T$ **do**
2 $\quad$ Compute empirical motif densities $\hat{v}(G_t) \in \mathbb{R}^m$ over $\mathcal{F}$.
3 Let $V \leftarrow [\hat{v}(G_1), \ldots, \hat{v}(G_T)]^{\top} \in \mathbb{R}^{T \times m}$.
4 Run $k$-means on $V$ with $K$ clusters to obtain clusters $\{C_1, \ldots, C_K\}$ and centroids $\{\mu_1, \ldots, \mu_K\}$.
5 Define assignment $\pi(G_t) \leftarrow \arg\min_{k \in \{1, \ldots, K\}} \|\hat{v}(G_t) - \mu_k\|_2$.
6 **for** $k = 1$ **to** $K$ **do**
7 $\quad$ Select $L$ graphs in $C_k$ closest to $\mu_k$ in moment space: $S_k \subseteq C_k$.
8 $\quad$ Estimate cluster graphon $\hat{W}_k$ using $S_k$ (e.g., via SIGL), and obtain latent positions $\{\eta_i\}$.
9 **return** $\{\hat{W}_k\}_{k=1}^{K}$ and $\pi$.

---

**Algorithm 2:** Graphon Mixture-Aware Mixup (GMAM)

**Input** : Labeled graphs $\mathcal{D} = \{(G_t, y_t)\}_{t=1}^{T}$, $y_t \in \{1, \ldots, C\}$; operator $\Phi$ (Alg. 1) run *per class*; mixing coefficient distribution $\lambda \sim \text{Uniform}(0, 0.2)$; target node count $n$; augmentation ratio $r \in (0, 1]$.

**Output:** Augmented set $\widetilde{\mathcal{D}} = \{(G_\lambda^{(m)}, \mathbf{y}_\lambda^{(m)})\}_{m=1}^{M}$ of size $M = \lceil rT \rceil$.

1 Partition data by class: $\mathcal{D}_i = \{G_t \mid y_t = i\}$ for $i = 1, \ldots, C$.
2 **for** $i = 1$ **to** $C$ **do**
3 $\quad$ $(\{\hat{W}_{i,1}, \ldots, \hat{W}_{i,K_i}\}, \pi_i) \leftarrow \Phi(\mathcal{D}_i)$.
4 $M \leftarrow \lceil rT \rceil$, $\quad \widetilde{\mathcal{D}} \leftarrow \varnothing$.
5 **for** $m = 1$ **to** $M$ **do**
6 $\quad$ Sample distinct classes $i \neq j$; sample graphs $G_a \in \mathcal{D}_i$, $G_b \in \mathcal{D}_j$.
7 $\quad$ Identify graphons: $\hat{W}_{i,a} \leftarrow \hat{W}_{i,\pi_i(G_a)}$, $\quad \hat{W}_{j,b} \leftarrow \hat{W}_{j,\pi_j(G_b)}$.
8 $\quad$ Sample $\lambda \sim \text{Uniform}(0, 0.2)$ and set $W_\lambda \leftarrow \lambda \hat{W}_{i,a} + (1 - \lambda)\hat{W}_{j,b}$.
9 $\quad$ Sample latent positions $u_1, \ldots, u_n \overset{\text{i.i.d.}}{\sim} \text{Uniform}[0, 1]$.
10 $\quad$ **for** $1 \leq p < q \leq n$ **do**
11 $\quad\quad$ Draw $A_\lambda(p, q) \sim \text{Bernoulli}(W_\lambda(u_p, u_q))$ and set $A_\lambda(q, p) \leftarrow A_\lambda(p, q)$.
12 $\quad$ Construct $G_\lambda^{(m)}$ from $A_\lambda$ (and optional node features).
13 $\quad$ Set $\mathbf{y}_\lambda^{(m)} \leftarrow \lambda \, \mathbf{e}_i + (1 - \lambda) \, \mathbf{e}_j$ (one-hot $\mathbf{e}_i$).
14 $\quad$ $\widetilde{\mathcal{D}} \leftarrow \widetilde{\mathcal{D}} \cup \{(G_\lambda^{(m)}, \mathbf{y}_\lambda^{(m)})\}$.
15 **return** $\widetilde{\mathcal{D}}$.

---

---

**Algorithm 3:** Model-Aware Graph Contrastive Learning (MGCL)

---

**Input** : Unlabeled graphs $\mathcal{D} = \{G_t\}_{t=1}^T$; shared encoder $\mathcal{E}_\theta$; operator $\Phi$ (Alg. 1); temperature $\tau$; batch size $L$; augmentation rate $r\%$.
**Output:** Trained encoder parameters $\theta$.

---

1   $(\{\hat{W}_1, \ldots, \hat{W}_K\}, \pi) \leftarrow \Phi(\mathcal{D})$ to obtain cluster assignments $C_t \leftarrow \pi(G_t)$.
2   **for** *epoch* $= 1, 2, \ldots$ **do**
3     Shuffle $\mathcal{D}$ and partition into mini-batches $\mathcal{B}$ of size $L$.
4     **for** *batch* $\mathcal{B} = \{G_t\}_{t=1}^L$ **do**
5       **foreach** $G_t$ *in* $\mathcal{B}$ **do**
6         Let $A_t$ (adjacency) and $X_t$ (features) be inputs.
          `// Graphon-aware augmentation using cluster graphon` $\hat{W}_{C_t}$
7         Sample a subset $E_{\text{sel}}$ of $r\%$ node pairs.
8         **foreach** $(i,j) \in E_{\text{sel}}$ **do**
9           Draw $\tilde{A}_t(i,j) \sim \text{Bernoulli}(\hat{W}_{C_t}(\eta_i, \eta_j))$; set $\tilde{A}_t(j,i) \leftarrow \tilde{A}_t(i,j)$.
10        Set $\tilde{A}_t(i,j) \leftarrow A_t(i,j)$ for all $(i,j) \notin E_{\text{sel}}$.
11        Compute embeddings $z_t \leftarrow \mathcal{E}_\theta(A_t, X_t)$ and $\tilde{z}_t \leftarrow \mathcal{E}_\theta(\tilde{A}_t, X_t)$.
      `// Model-aware InfoNCE: negatives only from different`
        `clusters`
12       Initialize $\mathcal{L}_{\text{batch}} \leftarrow 0$.
13       **for** $t = 1$ **to** $L$ **do**
14         Define negative index set $\mathcal{N}_t \leftarrow \{t' \in \{1, \ldots, L\} \mid C_{t'} \neq C_t\}$.
15         **if** $\mathcal{N}_t = \varnothing$ **then**
16           **continue** (skip or resample batch).
17

$$\ell_t \leftarrow -\log \frac{\exp(\text{sim}(z_t, \tilde{z}_t)/\tau)}{\sum\limits_{t' \in \mathcal{N}_t} \exp(\text{sim}(z_t, \tilde{z}_{t'})/\tau)}$$

        $\mathcal{L}_{\text{batch}} \leftarrow \mathcal{L}_{\text{batch}} + \ell_t$.
18       $\mathcal{L}_{\text{batch}} \leftarrow \mathcal{L}_{\text{batch}}/|\{t : \mathcal{N}_t \neq \varnothing\}|$.
19       Update $\theta$ via gradient step on $\mathcal{L}_{\text{batch}}$.
20 **return** $\theta$.

---

## A.1   COMPLEXITY ANALYSIS OF ALGORITHM 1

The overall time complexity $T_\Phi$ of the algorithm is the sum of its three constituent stages: motif counting, k-means clustering, and per-cluster graphon estimation. The combined complexity is given by:

$$T_\Phi = O\left(\underbrace{T(e_{\max}d_{\max} + n_{\max}d_{\max}^3)}_{\text{Motif Counting}} + \underbrace{I \cdot T \cdot m \log T}_{\text{k-means}} + \underbrace{L \cdot N_e \cdot n_{\max}^2 \log T}_{\text{Graphon Estimation}}\right)$$

Each term in this expression corresponds to a distinct phase of the algorithm. The first term, $O(T(e_{\max}d_{\max} + n_{\max}d_{\max}^3))$, represents the cost of computing the densities for all 9 motifs of size 4 across $T$ graphs, using an efficient graphlet counting algorithm like ORCA (Hočevar & Demšar, 2014). While this term's theoretical worst-case for dense graphs is $O(T \cdot n_{\max}^4)$, making it the apparent bottleneck, our own empirical analysis demonstrates a practical runtime that scales closer to cubic, $O(T \cdot n_{\max}^3)$. This significant practical speed-up is due to an **extremely small leading constant** ($c \approx 2.97 \times 10^{-8}$), a key finding that underpins our method's efficiency (Ramezanpour et al., 2025).

The second term, $O(I \cdot T \cdot m \log T)$, is the standard and well-established complexity for k-means clustering (e.g., via Lloyd's algorithm), which partitions the $T$ motif vectors in $\mathbb{R}^m$ into $K = \log T$ clusters over $I$ iterations. Finally, the third term, $O(L \cdot N_e \cdot n_{\max}^2 \log T)$, accounts for the per-cluster

refinement stage. Here, a powerful graphon estimator, such as SIGL (Azizpour et al., 2025), is executed for each of the $K$ clusters on a representative subset of $L$ graphs, requiring $N_e$ training epochs.

In conclusion, while the theoretical complexity suggests motif counting is the most expensive step, its low empirical constant makes the entire pipeline computationally feasible and scalable. This practical efficiency allows our method to effectively operate on many smaller, subsampled graphs, which is a core aspect of its design.

## A.2 RUNTIME ANALYSIS

In this section, we compute three runtime components for both GMAM and MGCL:

1. The time required to compute motif vectors for all graphs in the dataset (identical for GMAM and MGCL);

2. The time required to perform the first step of our method, which includes clustering and graphon estimation. This step is applied per class in GMAM and once for the entire dataset in MGCL, using the default number of clusters;

3. The time required for actual training and validation, which is independent of the previous steps—that is, motif computation and graphon estimation do not affect training time.

Figure 3 reports these three runtimes for each dataset and setup. As shown, the time required for motif computation, clustering, and graphon estimation is consistently much smaller than the training time, with the gap being especially pronounced in datasets containing a larger number of graphs, such as COLLAB, NCI1, and REDDIT-MULTI-5K.

Combined with the results in Section 4, which show that our method outperforms existing state-of-the-art approaches, the additional cost required for motif-based clustering is well justified.

Finally, we note that in these experiments, we use SIGL for graphon estimation. If runtime were of primary importance, one could instead employ faster estimators such as USVT or SAS.

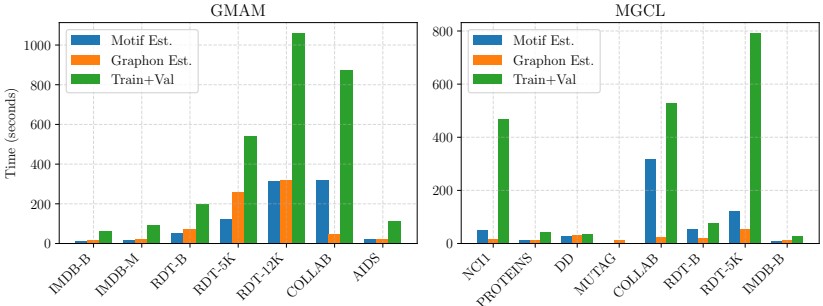

Figure 3: Runtime analysis comparing the time required for motif vector computation, graphon estimation, and model training. (a) GMAM (b) MGCL

## B TECHNICAL PROOFS

This appendix provides the proof for Theorem 1 and compares our novel error bound to the classical approach.

### B.1 PROOF OF THEOREM 1

The proof relies on bounding the difference between the empirical densities using the triangle inequality:

$$|t(F, G_1) - t(F, G_2)| \leq \underbrace{|t(F, W_1) - t(F, W_2)|}_{\text{Graphon Difference}} + \underbrace{|t(F, G_1) - t(F, W_1)|}_{\text{Sampling Error 1}} + \underbrace{|t(F, G_2) - t(F, W_2)|}_{\text{Sampling Error 2}}.$$

The **Graphon Difference** term is bounded by the Counting Lemma (cf. Theorem 10.23 in Lovász (2012)):

$$|t(F, W_1) - t(F, W_2)| \leq e(F)d_{\text{cut}}(W_1, W_2) \leq e(F)\epsilon.$$

The **Sampling Error** terms, $|t(F, G_i) - t(F, W_i)|$, are bounded using a novel two-stage analysis that separates the randomness from vertex selection and edge realization. This approach yields a tighter bound than classical single-stage methods. We decompose the total sampling error into two components:

$$|t(F, G) - t(F, W)| \leq \underbrace{|\widehat{t}(F; X) - t(F, W)|}_{\text{Vertex Noise}} + \underbrace{|t(F, G) - \widehat{t}(F; X)|}_{\text{Edge Noise}}$$

where $\widehat{t}(F; X)$ is the conditional expectation of the density given the sampled vertex labels $X = (X_1, \ldots, X_n)$. The following two lemmas bound the vertex and edge noise, respectively.

**Lemma 1** (Bounding Vertex Noise). *Let $m = \lfloor n/k \rfloor$. For any $\delta_v > 0$,*

$$\Pr\left[|\widehat{t}(F; X) - t(F, W)| \geq \delta_v\right] \leq 2\exp\left(-2m\,\delta_v^2\right).$$

**Lemma 2** (Bounding Edge Noise). *Conditioned on the vertex labels $X = (X_i)_{i=1}^n$, for any $\delta_e > 0$,*

$$\Pr\left[|t(F, G) - \widehat{t}(F; X)| \geq \delta_e \mid X\right] \leq 2\exp\left(-\frac{n(n-1)\,\delta_e^2}{2e(F)^2}\right).$$

**Combined Bound** To obtain the total sampling error for one graph, $|t(F, G_i) - t(F, W_i)|$, with a target failure probability $\eta$, we allocate $\eta/2$ to each noise source (vertex and edge). Inverting the bounds in the lemmas above, we set:

$$\delta_v = \sqrt{\frac{1}{2m}\log\frac{4}{\eta}} \quad \text{and} \quad \delta_e = \frac{e(F)}{\sqrt{n(n-1)}}\sqrt{2\log\frac{4}{\eta}}.$$

The total sampling error for one graph is bounded by $\delta_s = \delta_v + \delta_e$ with probability at least $1 - \eta$.

Applying this to both sampling error terms ($|t(F, G_1) - t(F, W_1)|$ and $|t(F, G_2) - t(F, W_2)|$) and using a union bound, we get the final result in Theorem 1 with probability at least $1 - 2\eta$:

$$|t(F, G_1) - t(F, G_2)| \leq e(F)\epsilon + 2(\delta_v + \delta_e) = e(F)\epsilon + 2\left(\sqrt{\frac{1}{2m}\log\frac{4}{\eta}} + \frac{e(F)}{\sqrt{n(n-1)}}\sqrt{2\log\frac{4}{\eta}}\right).$$

## B.2 COMPARISON WITH THE CLASSICAL BOUND

To highlight the improvement offered by our two-stage approach, we now present the classical single-stage bound and compare them.

### B.2.1 THE CLASSICAL SINGLE-STAGE BOUND

The standard approach (cf. Lemma 4.4 in Borgs et al. (2008)) treats the empirical motif density $t(F, G)$ as a complex function of $n$ random variables and applies McDiarmid's inequality directly. This results in the following concentration result.

**Lemma 3** (Classical Concentration of Motif Densities). *Let $F$ be a fixed graph with $k = v(F)$ vertices. Let $G$ be an $n$-vertex graph sampled from a graphon $W$. Then for any $\delta > 0$,*

$$\Pr[|t(F, G) - t(F, W)| \geq \delta] \leq 2\exp\left(-\frac{n\delta^2}{4k^2}\right).$$

The limitation of this approach is that changing a single vertex can alter many edge probabilities, leading to a loose bound with a factor of $k^2$ in the exponent.

### B.2.2 Asymptotic Comparison

We now compare the sampling error from the classical approach with our novel bound. For a total failure probability $\eta$:

- **Classical Bound:** Inverting the bound in Lemma 3 yields:

$$\delta_s^{\text{old}}(\eta) = 2k\sqrt{\frac{1}{n}\log\frac{2}{\eta}}$$

- **Novel Bound:** Approximating $m \approx n/k$ and $n(n-1) \approx n^2$, we have:

$$\delta_s^{\text{new}}(\eta) \approx \left(\frac{\sqrt{k}}{\sqrt{2n}} + \frac{e(F)\sqrt{2}}{n}\right)\sqrt{\log\frac{4}{\eta}}$$

The crucial difference lies in their dependence on the motif size, $k$. The classical bound $\delta_s^{\text{old}}$ scales linearly with $k$ ($O(k)$), whereas the dominant term in our novel bound $\delta_s^{\text{new}}$ scales with the square root of $k$ ($O(\sqrt{k})$). As $\sqrt{k}$ grows much more slowly than $k$, our novel bound is substantially tighter for larger motifs.

### B.2.3 Numerical Validation

We validate this theoretical improvement by computing both bounds for all motifs with up to 4 vertices ($k \leq 4$), for graph sizes $n$ from 50 to 1000, at a 95% confidence level ($\eta = 0.05$). The results are visualized in Figure 4. The novel bound is uniformly tighter than the classical bound across all tested motifs.

## C  Proof of Theorem 2

**Notation.** Let $\mathbf{z}_t = \mathcal{E}(A_t, X_t)$ denote the embedding of graph $t$ obtained by applying the encoder $\mathcal{E}$ to its adjacency matrix $A_t$ and feature matrix $X_t$. Let $\text{sim}(u, v)$ be cosine similarity and define $\theta(u, v) := \frac{\text{sim}(u,v)}{\tau}$. For an anchor graph $t$, write the set of all negatives in standard InfoNCE as $\mathcal{N}_t := \{\, k \in \{1, \ldots, L\} \setminus \{t\} \,\}$, and the *cluster-restricted* negatives in our case as $\widetilde{\mathcal{N}}_t := \{\, k \in \mathcal{N}_t : C_k \neq C_t \,\}$, where $C_i$ is the cluster of graph $i$. For brevity, let $s_k := \theta(\mathbf{z}_t, \mathbf{z}'_k)$ and $\tilde{s}_k := \theta(\mathbf{z}_t, \tilde{\mathbf{z}}_k)$ where $\mathbf{z}'_t$ and $\tilde{\mathbf{z}}_t$ are the embedding of the positive sample obtained from a random and graphon-consistent augmentation, respectively. The standard InfoNCE loss for graph $t$ is defined as

$$\ell_{\text{InfoNCE}}(t) = -\log\frac{\exp(\theta(\mathbf{z}_t, \mathbf{z}'_t))}{\sum_{k \in \mathcal{N}_t}\exp(s_k)}. \tag{18}$$

Our proposed cluster-restricted loss is defined as

$$\ell_{\text{cluster}}(t) = -\log\frac{\exp(\theta(\mathbf{z}_t, \tilde{\mathbf{z}}_t))}{\sum_{k \in \widetilde{\mathcal{N}}_t}\exp(\tilde{s}_k)}. \tag{19}$$

Using the natural logarithm and separating the positive-pair term, we can rewrite:

$$\ell_{\text{InfoNCE}}(t) = -\theta(\mathbf{z}_t, \mathbf{z}'_t) + \ln\left(\sum_{k \in \mathcal{N}_t} e^{s_k}\right), \tag{20}$$

$$\ell_{\text{cluster}}(t) = -\theta(\mathbf{z}_t, \tilde{\mathbf{z}}_t) + \ln\left(\sum_{k \in \widetilde{\mathcal{N}}_t} e^{\tilde{s}_k}\right). \tag{21}$$

**Proposition 1** (Lower bound for the cluster-restricted loss)**.** *Let*

$$\ell_{cluster}(t) = -\theta(\mathbf{z}_t, \tilde{\mathbf{z}}_t) + \ln\sum_{k \in \widetilde{\mathcal{N}}_t}\exp\big(\theta(\mathbf{z}_t, \mathbf{z}'_k)\big),$$

*where $\widetilde{\mathcal{N}}_t = \{\, k \neq t : C_k \neq C_t \,\}$ and $m_t := |\widetilde{\mathcal{N}}_t|$. Then, for every anchor $t$,*

$$\ell_{cluster}(t) \;\geq\; -\theta(\mathbf{z}_t, \tilde{\mathbf{z}}_t) \;+\; \frac{1}{m_t}\sum_{k \in \widetilde{\mathcal{N}}_t}\theta(\mathbf{z}_t, \mathbf{z}'_k) \;+\; \ln m_t. \tag{22}$$

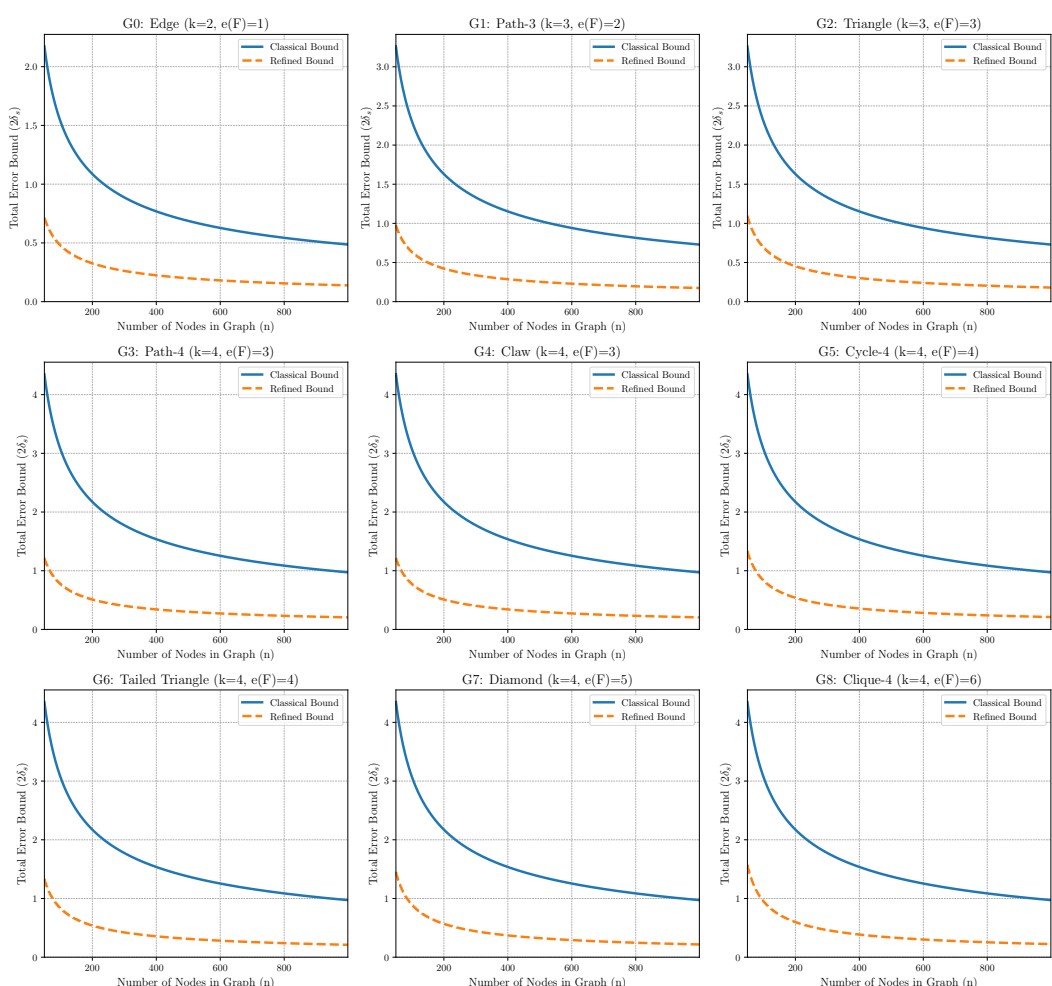

Figure 4: Comparison of the total error bounds ($2\delta_s$) for the classical (solid blue) and novel (dashed orange) approaches. The novel bound is consistently tighter, with the gap widening for motifs with more vertices ($k$), confirming its superior $O(\sqrt{k})$ scaling.

*Proof.* By definition,

$$\ln \sum_{k \in \widetilde{\mathcal{N}}_t} \exp\big(\theta(\mathbf{z}_t, \mathbf{z}'_k)\big) = \ln m_t + \ln\left(\frac{1}{m_t} \sum_{k \in \widetilde{\mathcal{N}}_t} \exp\big(\theta(\mathbf{z}_t, \mathbf{z}'_k)\big)\right).$$

Applying Jensen's inequality to the convex exponential function yields

$$\ln \sum_{k \in \widetilde{\mathcal{N}}_t} \exp\big(\theta(\mathbf{z}_t, \mathbf{z}'_k)\big) \;\geq\; \ln m_t + \frac{1}{m_t} \sum_{k \in \widetilde{\mathcal{N}}_t} \theta(\mathbf{z}_t, \mathbf{z}'_k).$$

Substituting into the definition of $\ell_{\text{cluster}}(t)$ gives the claimed lower bound. $\square$

**Proposition 2** (Lower bound for the standard InfoNCE loss)**.** *Let*

$$\ell_{InfoNCE}(t) = -\theta(\mathbf{z}_t, \mathbf{z}'_t) + \ln \sum_{k \in \mathcal{N}_t} \exp\big(\theta(\mathbf{z}_t, \mathbf{z}'_k)\big),$$

*where* $\mathcal{N}_t = \{\, k \neq t \,\}$ *and* $n_t := |\mathcal{N}_t| = N - 1$. *Define the negative-center score*

$$\overline{\theta}_{-t} := \frac{1}{n_t} \sum_{k \in \mathcal{N}_t} \theta(\mathbf{z}_t, \mathbf{z}'_k).$$

*Then, for every anchor $t$,*

$$\ell_{\textit{InfoNCE}}(t) \;\geq\; -\theta(\mathbf{z}_t, \mathbf{z}_t') \;+\; \overline{\theta}_{-t} \;+\; \ln n_t. \tag{23}$$

*Proof.* Apply Jensen's inequality to the convex exponential:

$$\ln \sum_{k \in \mathcal{N}_t} e^{\theta(\mathbf{z}_t, \mathbf{z}_k')} = \ln n_t + \ln\left(\tfrac{1}{n_t} \sum_{k \in \mathcal{N}_t} e^{\theta(\mathbf{z}_t, \mathbf{z}_k')}\right) \;\geq\; \ln n_t + \tfrac{1}{n_t} \sum_{k \in \mathcal{N}_t} \theta(\mathbf{z}_t, \mathbf{z}_k') = \ln n_t + \overline{\theta}_{-t}.$$

Substitute into the definition of $\ell_{\text{InfoNCE}}(t)$ to obtain equation 23. $\qquad\square$

As a result, as discussed in Section 3.3 and shown by the above proofs together with Proposition 2, minimizing our refined loss function pushes each graph away from the centroid of graphs generated by other models, whereas the standard InfoNCE loss uses all graphs and pushes it away from the centroid of the entire dataset.

## D  GRAPHON ESTIMATION

Here we expalin the details of SIGL. The goal is to estimate an unknown graphon $\omega : [0,1]^2 \to [0,1]$, given a set of graphs $\mathcal{D} = \{\mathbf{A}_t\}_{t=1}^M$ sampled from it. Since using the Gromov-Wasserstein (GW) Mémoli (2011) distance is computationally infeasible for large graphs, the SIGL framework Azizpour et al. (2025) proposes a scalable three-step procedure:

**Step 1: Sorting nodes via latent variable estimation**  To align all graphs to a common node ordering (which is crucial for consistent estimation), SIGL estimates latent variables $\hat{\boldsymbol{\eta}}_t = \{\hat{\eta}_i\}_{i=1}^{n_t}$ for each graph $G_t$ using a Graph Neural Network (GNN):

$$\hat{\boldsymbol{\eta}}_t = g_{\phi_1}(\mathbf{A}_t, \mathbf{Y}_t), \quad \text{where } \mathbf{Y}_t \sim \mathcal{N}(0,1)$$

An auxiliary graphon $h_{\phi_2}$ modeled by an Implicit Neural Representation (INR) maps pairs of latent variables to edge probabilities:

$$h_{\phi_2}(\hat{\boldsymbol{\eta}}_t(i), \hat{\boldsymbol{\eta}}_t(j)) \approx \mathbf{A}_t(i,j)$$

The latent variables and auxiliary graphon are jointly trained by minimizing the mean squared error to get $\phi = \{\phi_1 \cup \phi_2\}$:

$$\mathcal{L}(\phi) = \sum_{t=1}^M \frac{1}{n_t^2} \sum_{i,j=1}^{n_t} \left[\mathbf{A}_t(i,j) - h_{\phi_2}(\hat{\boldsymbol{\eta}}_t(i), \hat{\boldsymbol{\eta}}_t(j))\right]^2$$

A sorting permutation $\hat{\pi}$ is defined based on the learned latent variables:

$$\hat{\boldsymbol{\eta}}_t(\hat{\pi}(1)) \geq \hat{\boldsymbol{\eta}}_t(\hat{\pi}(2)) \geq \cdots \geq \hat{\boldsymbol{\eta}}_t(\hat{\pi}(n_t))$$

In a nutshell, this permutation sorts the latent variables from $0$ to $1$. The graphs are reordered accordingly to produce sorted adjacency matrices $\hat{\mathbf{A}}_t$.

**Step 2: Histogram Approximation**  For each sorted graph $\hat{\mathbf{A}}_t$, a histogram $\hat{\mathbf{H}}_t \in \mathbb{R}^{k \times k}$ is computed using average pooling with window size $h$:

$$\hat{\mathbf{H}}_t(i,j) = \frac{1}{h^2} \sum_{s_1=1}^{h} \sum_{s_2=1}^{h} \hat{\mathbf{A}}_t\left((i-1)h + s_1, (j-1)h + s_2\right)$$

This results in a new dataset $\mathcal{I} = \{\hat{\mathbf{H}}_t\}_{t=1}^M$, providing discrete, noisy views of the unknown graphon $\omega$.

**Step 3: Training the Graphon INR** The final step constructs a supervised dataset $\mathcal{C}$ from all histograms, where each point corresponds to a coordinate-value triple:

$$\mathcal{C} = \left\{ \left( \frac{i}{k_t}, \frac{j}{k_t}, \hat{\mathbf{H}}_t(i,j) \right) : i,j \in \{1, \ldots, k_t\}, t \in \{1, \ldots, M\} \right\}$$

A second INR structure $f_\theta : [0,1]^2 \to [0,1]$ is then trained to regress the graphon values by minimizing the MSE:

$$\mathcal{L}(\theta) = \sum_{(x,y,z) \in \mathcal{C}} (f_\theta(x,y) - z)^2$$

This scalable approach enables two things: 1) the estimation of a continuous graphon $\omega$ using large-scale graph data without relying on costly combinatorial metrics like the GW distance, and 2) the estimation of the latent variables given an input graphs, i.e., an inverse mapping $\mathcal{W}^{-1} : \mathbf{A} \to \boldsymbol{\eta}$.

# E EXPERIMENTAL DETAILS.

## E.1 GROUND TRUTH GRAPHONS

In Table 4, we provide the mathematical definition of the graphon used in Section 4 for the synthetic experiments.

Table 4: Ground truth graphons.

| | $\omega(x,y)$ |
|---|---|
| 0 | $xy$ |
| 1 | $\exp(-(x^{0.7} + y^{0.7}))$ |
| 2 | $\frac{1}{4}(x^2 + y^2 + \sqrt{x} + \sqrt{y})$ |
| 3 | $\frac{1}{2}(x + y)$ |
| 4 | $(1 + \exp(-2(x^2 + y^2)))^{-1}$ |
| 5 | $(1 + \exp(-\max\{x,y\}^2 - \min\{x,y\}^4))^{-1}$ |
| 6 | $\exp(-\max\{x,y\}^{0.75})$ |

## E.2 HYPER-PARAMETERS

**Graphon estimation** The hyperparameters used to estimate the graphon with SIGL across its three steps, as described in the previous section, are as follows for both mixup and graph level tasks. We use the `Adam` optimizer Kingma (2015) with a learning rate of $lr = 0.01$ for both Step 1 and Step 3, running for 40 and 20 epochs, respectively. In Step 1, the batch size is set to 1 graph, while in Step 3, each batch includes 1024 data points from $\mathcal{C}$. In Step 1, the GNN, $g_{\phi_1}$ comprises two consecutive graph convolutional layers, each followed by a ReLU activation function. All convolutional layers use 8 hidden channels. The INR structures in Step 1 ($h_{\phi_2}$) and Step 3 ($f_\theta$) each have 3 layers with 20 hidden units per layer and use a default frequency of 10 for the $\sin(.)$ activation function.

**Mixup** To ensure a fair comparison, we use the same hyperparameters for model training and the same architecture across vanilla models and other baselines. Also, we conduct the experiments using the same hyperparameter values as in Han et al. (2022). For graph classification tasks, we employ the `Adam` optimizer with an initial learning rate of 0.01, which is halved every 100 epochs over a total of 800 epochs. The batch size is set to 128. The dataset is split into training, validation, and test sets in a 7:1:2 ratio. The best test epoch is selected based on validation performance, and test accuracy is reported over eight runs with the same `seed` used in Han et al. (2022).

We generate 20% more graphs for training. The graphons are estimated from the training graphs, and we use different $\lambda$ values in the range $[0.1, 0.2]$ to control the strength of mixing in the generated synthetic graphs. For graphon estimation using SIGL and IGNR, we use 20% of the training data per class to estimate the graphon. The new graphs are generated with the average number of nodes as defined for the primary G-Mixup case, which is identified as the optimal size. All methods are evaluated using the same random seeds.

**MGCL**  To train the encoder $\mathcal{E}_\theta$, we follow the configuration from GraphCL You et al. (2020). We use the Adam optimizer with a learning rate of $lr = 0.001$, training the encoder for 20 epochs. The encoder is implemented as a 3-layer GIN Xu et al. (2019) network, with each layer consisting of 32 hidden units followed by a ReLU activation. A final linear projection head maps the output to a 32-dimensional graph-level representation. For a fair comparison, we set the resampling ratio to $r = 20\%$, consistent with other methods. All methods are evaluated using the same random seeds.

### E.3  DATASET DETAILS

In Table 5, we report the statistics of all datasets from TUDataset (Morris et al., 2020) used in our real-world experiments.

Table 5: Benchmark datasets statistics.

| Statistic | Biochemical Molecules | | | | | Social Networks | | | |
|---|---|---|---|---|---|---|---|---|---|
| | NCI1 | PROTEINS | DD | MUTAG | AIDS | COLLAB | RDT-B | RDT-M5K | IMDB-B |
| #Graphs | 4,110 | 1,113 | 1,178 | 188 | 2,000 | 5,000 | 2,000 | 4,999 | 1,000 |
| Avg. #Nodes | 29.87 | 39.06 | 284.32 | 17.93 | 15.69 | 74.5 | 429.6 | 508.8 | 19.8 |
| Avg. #Edges | 32.30 | 72.82 | 715.66 | 19.79 | 16.20 | 2457.78 | 497.75 | 594.87 | 96.53 |
| #Classes | 2 | 2 | 2 | 2 | 2 | 3 | 2 | 5 | 2 |

Note that several graph-level datasets lack explicit node attributes. In such cases, one-hot encoding of node degrees is commonly used to construct node features.

### E.4  BASELINE MODELS.

Here we have a small description of the baselines used mixup and GCL experiments.

**Mixup baselines.**

- *Vanilla*, a baseline with no augmentation;
- *DropEdge* (Rong et al., 2019), which uniformly removes a fraction of edges;
- *DropNode* (Feng et al., 2020), which randomly drops a portion of nodes;
- *Subgraph* (You et al., 2020), which extracts random-walk-based subgraphs;
- *M-Mixup* (Wang et al., 2021), which linearly interpolates graph-level representations;
- *SubMix* (Yoo et al., 2022), which mixes random subgraphs of graph pairs;
- *G-Mixup* (Han et al., 2022), which performs class-level graph Mixup by interpolating graphons from different classes;
- *S-Mixup* (Ling et al., 2023), which employs soft alignment for graph interpolation;
- *SIGL* (Azizpour et al., 2025), which replaces the original graphon estimator in G-Mixup with SIGL; and
- *MomentMixup* (Ramezanpour et al., 2025), which mixes motif moment vectors and generates new samples from the mixed vector.

**GCL baselines.**

- InfoGraph Sun et al. (2020): A variant of DGI (Veličković et al., 2019) that maximizes mutual information between graph-level and substructure representations at multiple scales.
- GraphCL You et al. (2020): Learns representations via predefined augmentations such as edge perturbation, node dropping, and attribute masking.
- MVGRL Hassani & Khasahmadi (2020): Performs contrastive learning between structural views, e.g., adjacency and diffusion representations.
- JOAO You et al. (2021): Uses min-max optimization to adaptively select augmentations during training.

|   | 1 | 2 | 3 | 4 | 5 | 6 |
|---|-------|-------|-------|-------|-------|-------|
| **0** | 0.133 | 0.265 | 0.264 | 0.530 | 0.418 | 0.271 |
| **1** |       | 0.180 | 0.189 | 0.433 | 0.308 | 0.173 |
| **2** |       |       | 0.024 | 0.270 | 0.175 | 0.111 |
| **3** |       |       |       | 0.275 | 0.190 | 0.126 |
| **4** |       |       |       |       | 0.139 | 0.284 |
| **5** |       |       |       |       |       | 0.162 |

Table 6: Pairwise GW distances between groundtruth graphons.

- Ad-GCL Suresh et al. (2021): An adversarial training–based GCL framework that introduces an edge-perturbation process designed as an attack to improve robustness.

- AutoGCL Yin et al. (2022): Employs learnable view generators with auto-augmentation for adaptive, label-preserving samples.

- SimGRACE Xia et al. (2022): Replaces graph augmentations with encoder-level noise to enforce consistency.

- RGCL Li et al. (2022): Generates rationale-aware views by identifying substructures most relevant for discrimination.

- DRGCL Ji et al. (2024): Learns dimensional rationales and applies bi-level meta-learning to mitigate confounding noise.

### E.5 GRAPH EMBEDDING BASELINE DETAILS

This section provides further details on the baseline graph embedding methods used for comparison in the synthetic clustering experiment (Section 4.1).

**GW distance of ground truth graphons.**

**Baseline Methods.** We compare our moment vectors against five widely-used graph representation methods. For methods that produce node-level embeddings (GCN, GIN, DeepWalk, Spectral), we obtain a graph-level embedding by applying a global mean-pooling operation over all node embeddings in the graph. The target embedding dimension for all baselines was set to 32.

- **Moment Embedding (Ours):** This is our proposed method, where each graph is represented by a 9-dimensional vector of its empirical motif densities for all connected motifs up to 4 nodes.

- **Graph Convolutional Network (GCN):** A popular Graph Neural Network (GNN) architecture that learns node representations by iteratively aggregating feature information from local neighborhoods Kipf & Welling (2016). As our clustering task is unsupervised, the GCN is not trained; we use a randomly initialized network to generate embeddings. This standard approach tests the intrinsic structural representation power of the architecture itself.

- **Graph Isomorphism Network (GIN):** A powerful GNN model proven to be as discriminative as the Weisfeiler-Leman test for graph isomorphism Xu et al. (2019). Similar to our use of GCN, the GIN model's weights are kept random, allowing it to serve as an unsupervised feature extractor without training on downstream labels.

- **Graph2Vec:** An unsupervised whole-graph embedding method inspired by natural language processing. It treats graphs as documents and rooted subgraphs as "words," learning representations that capture structural similarities between graphs Narayanan et al. (2017).

- **DeepWalk:** A pioneering node embedding technique that learns latent representations by applying language modeling techniques (Skip-Gram) to sequences of nodes generated from truncated random walks on the graph Perozzi et al. (2014).

- **Spectral Embedding:** A classical approach that utilizes the eigenvectors of the graph's Laplacian matrix. The first few non-trivial eigenvectors form a low-dimensional representation of the nodes, capturing the graph's global connectivity structure Ng et al. (2001).

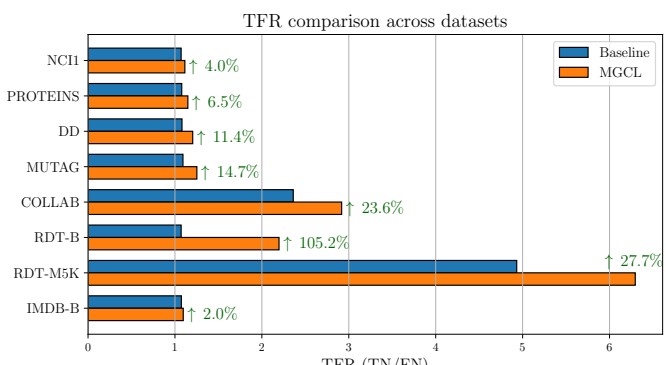

Figure 5: Effect of clustering on TFR across different datasets.

### E.6 COMPUTE RESOURCES

All experiments were conducted on a server running Ubuntu 20.04.6 LTS, equipped with an AMD EPYC 7742 64-Core Processor and an NVIDIA A100-SXM4-80GB GPU with 80GB of memory. For model development, we utilized PyTorch version 1.13.1, along with PyTorch Geometric version 2.3.1, which also served as the source for all datasets used in our study.

## F ADDITIONAL EXPERIMENTS.

**False negative reduction.** To evaluate the effect of clustering on the rate of false negatives, we define the True Negative to False Negative Ratio (TFR).

To compute this metric, in each data batch, we examine the negative samples relative to a graph $i$. Among these, the negative samples that share the same class as graph $i$ are considered false negatives, while those with a different class are treated as true negatives. Mathematically, TFR is defined as TFR $= \frac{1}{B} \sum_{i=1}^{B} \frac{|\mathcal{TN}(i)|}{\max\{1, |\mathcal{FN}(i)|\}}$. Note that in InfoNCE-based methods, all graphs in the batch except graph $i$ itself are treated as negative samples. However, in MGCL, the set of negative samples is smaller, as we exclude graphs from the same cluster as $i$. Although this reduced set naturally results in fewer false negatives, computing the relative ratio of true negatives to false negatives (TFR) ensures a fair comparison across methods. We compute the TFR for each graph in the batch and then average it across all graphs in the dataset. As shown in Figure 5, this metric increases across all datasets compared to the baseline (which represents all InfoNCE-based methods). This also provides evidence that motif-based clustering helps uncover the true underlying structure of the data.

### F.1 ABLATION STUDY ON NUMBER OF MOTIFS

To justify our choice of using motifs up to 4 nodes (a 9-dimensional feature vector) in the main experiment, we conduct an ablation study to analyze the effect of the number of motifs on clustering performance. We extend the experimental setup from Section 4.1 by computing densities for all 30 connected motifs with up to 5 nodes. We then iteratively evaluate the clustering accuracy of our methods by using an increasing number of motifs.

The results for the first 15 motifs are presented in Table 7. A clear trend emerges: accuracy rises sharply with the inclusion of the first few motifs, but then exhibits a long plateau of nearly constant performance. This indicates that some motifs are significantly more important for classification than others. A particularly notable increase occurs between $k = 8$ and $k = 9$ motifs in the *Varying* size setting, where accuracy jumps from 80.0% to 81.4% and then flatlines. This specific jump highlights the discriminative power of the 9th motif (the 4-cycle) and suggests that this initial set of motifs captures the most critical structural differences between the graphon families.

This study confirms that the initial 9 motifs provide the vast majority of the useful information. The minimal performance gain from adding more motifs (from $k = 10$ to $k = 15$ and beyond) does not justify the increased computational cost. This validates our use of the 9-dimensional moment vector in the main paper as an effective and efficient choice.

Table 7: Results of the ablation study for the first 15 motifs. Clustering accuracy (%) is shown for K-Means and Theory-Based methods across two dataset settings. The performance largely saturates after $k = 9$.

| # Motifs | Varying Size Accuracy (%) | | Fixed Size Accuracy (%) | |
|---|---|---|---|---|
| ($k$) | K-Means | Theory-Based | K-Means | Theory-Based |
| 1 | 70.0 | 74.3 | 74.7 | 74.7 |
| 2 | 67.1 | 77.1 | 74.7 | 76.9 |
| 3 | 78.6 | 80.0 | 79.1 | 82.3 |
| 4 | 78.6 | 80.0 | 79.1 | 82.7 |
| 5 | 78.6 | 80.0 | 79.1 | 82.9 |
| 6 | 78.6 | 80.0 | 79.1 | 83.0 |
| 7 | 78.6 | 80.0 | 79.1 | 83.0 |
| 8 | 78.6 | 80.0 | 79.1 | 83.0 |
| 9 | 80.0 | 81.4 | 79.4 | 83.6 |
| 10 | 80.0 | 81.4 | 79.4 | 83.6 |
| 11 | 80.0 | 81.4 | 79.4 | 83.6 |
| 12 | 80.0 | 81.4 | 79.4 | 83.6 |
| 13 | 80.0 | 81.4 | 79.4 | 83.6 |
| 14 | 80.0 | 81.4 | 79.4 | 83.6 |
| 15 | 80.0 | 81.4 | 79.4 | 83.6 |

## F.2 ABLATION STUDY ON NUMBER OF CLUSTERS.

### F.2.1 STRUCTURAL RECOVERY OF MIXTURES (MOMENT EMBEDDINGS)

To assess the quality of the clusters found by our approach and to study the effect of tuning the number of mixture components, $k$, we conducted an ablation study on our synthetic dataset, which has 7 ground-truth underlying mixtures. We measured the cluster quality using the **Adjusted Rand Index (ARI)**, where a higher score indicates a better match to the ground truth partitioning.

Table 8 (now **Table 10** in the paper) shows this comparison for our Moment Embeddings and the GCN baseline across various values of $k$ (from 2 to 10). We omitted other baselines for space, as GCN was the strongest performing baseline, with others having significantly lower ARI scores.

The results show that our Moment Embedding's ARI score **peaks at** $k = 6$ for both the varying-size and fixed-size graph datasets. This result is consistent with the ground truth of 7 mixtures. The peak at $k = 6$ rather than $k = 7$ is expected because the ground-truth Graphons 2 and 3 are structurally very similar (Gromov-Wasserstein distance of $0.024$, as noted in Section 4.1), and our method successfully groups these similar structures.

In contrast, the GCN baseline's cluster quality peaks at $k = 5$, and its peak ARI (e.g., $0.5709$ for fixed size) is significantly lower than our method's peak (e.g., $0.7166$ for varying size). This demonstrates that the GCN method fails to accurately recover the underlying graph structure.

### F.2.2 IMPACT ON DOWNSTREAM TASK PERFORMANCE

Here we evaluate the number of clusters $k$ on two downstream tasks: Multiple Graph Contrastive Learning (MGCL) and Graph Mixture Anomaly Mining (GMAM), using real-world datasets.

**MGCL** As mentioned in the Methods section, in order to match the size of the data, we use $K = \log T$ for the operator $\Phi$. Here we evaluate this number on three datasets with within the MGCL framework. In this section, we vary the number of clusters to evaluate its impact on overall performance.

Table 8: Adjusted Rand Index (ARI) vs. Number of Clusters ($k$) for all datasets.

| mode | Varying Size | | Fixed Size | |
|---|---|---|---|---|
| method | Moment Emb. | GCN Emb. | Moment Emb. | GCN Emb. |
| $k$ | | | | |
| 2 | 0.2048 | 0.2555 | 0.2048 | 0.2578 |
| 3 | 0.3616 | 0.3860 | 0.3616 | 0.4316 |
| 4 | 0.5463 | 0.4498 | 0.5463 | 0.4993 |
| 5 | 0.6296 | 0.4764 | 0.6223 | **0.5709** |
| 6 | **0.7166** | 0.4242 | **0.6889** | 0.5133 |
| 7 | 0.6709 | 0.3981 | 0.6444 | 0.4842 |
| 8 | 0.6650 | 0.4215 | 0.6756 | 0.4599 |
| 9 | 0.6276 | 0.4189 | 0.6463 | 0.4256 |
| 10 | 0.6953 | 0.4158 | 0.6073 | 0.4076 |

We vary the number of clusters from 1 to 10. For each value, we repeat the MGCL experiment across the same 10 trials using identical data partitions and report the average accuracy.

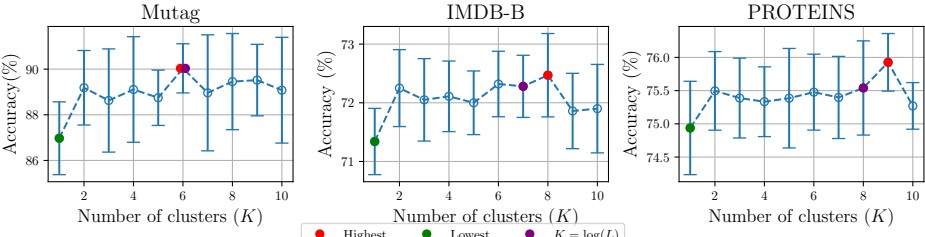

Figure 6: Effect of the number of clusters on MGCL performance.

The results are presented in Figure 6 for the MUTAG, and IMDB-BINARY datasets. A key observation is that using a single cluster—i.e., estimating one graphon for the entire dataset—leads to a drop in performance across all three datasets. In the IMDB-BINARY dataset, using 8 clusters yields better performance than the default setting of 7 clusters. However, using between 6 and 8 clusters consistently results in an average accuracy above 72.0%, suggesting that this range reasonably approximates the number of underlying models. A similar trend is observed in the PROTEINS dataset, where 9 clusters yield better performance than 8. For the MUTAG dataset, the highest average accuracy is achieved with the default setting of 6 clusters. These findings highlight that estimating multiple models improves performance. Still, the number of clusters remains a hyperparameter that should be tuned based on the dataset's characteristics, such as its variability and heterogeneity.

**GMAM**  We conduct a similar ablation in the GMAM setting by varying the number of clusters used to estimate the underlying models for each graph class. As shown in Figure 7, using more than one cluster consistently improves performance on both IMDB-B and IMDB-MULTI.

For IMDB-B, selecting between 2 and 9 clusters consistently outperforms the single-cluster setting, with performance peaking at the default value of 6 clusters. In IMDB-MULTI, using 2–10 clusters always matches or exceeds the accuracy obtained with a single cluster, and the best performance occurs at 4 clusters—slightly lower than the default of 6—suggesting that 4 clusters adequately capture the underlying generative models for this dataset.

In both datasets, using 10 clusters results in a slight decrease in accuracy, indicating that this number may be larger than necessary and could lead to over-fragmentation of the data, thereby overlooking shared structural patterns among graphs.

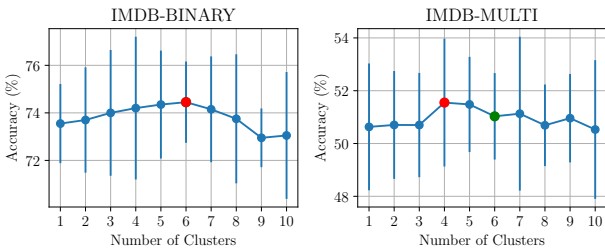

Figure 7: Effect of the number of clusters on GMAM performance.

### F.3 ABLATION ON THE NUMBER OF MIXTURE COMPONENTS

To thoroughly analyze the robustness of our mixture estimation, we conducted an ablation study on the number of underlying mixture components.

We constructed synthetic datasets with an increasing number of underlying mixtures, ranging from $k = 2$ to $k = 7$. We evaluated the performance using both the Adjusted Rand Index (ARI) and clustering accuracy. To ensure clarity, we present the ablation results in two separate tables: one for *varying-size graphs* (Table 9) and one for *fixed-size graphs* (Table 10).

Table 9: Ablation on Number of Mixtures (Varying Size).

| Num. Mixtures | ARI | | Accuracy | |
|---|---|---|---|---|
| | Moment Emb. | GCN | Moment Emb. | GCN |
| 2 | 1.0000 | 0.6383 | 1.0000 | 0.9000 |
| 3 | 1.0000 | 0.6633 | 1.0000 | 0.8667 |
| 4 | 0.6725 | 0.4707 | 0.7750 | 0.6500 |
| 5 | 0.7539 | 0.5977 | 0.8200 | 0.7200 |
| 6 | 0.8027 | 0.5517 | 0.8500 | 0.6833 |
| 7 | 0.7166 | 0.4375 | 0.8000 | 0.5857 |

Table 10: Ablation on Number of Mixtures (Fixed Size).

| Num. Mixtures | ARI | | Accuracy | |
|---|---|---|---|---|
| | Moment Emb. | GCN | Moment Emb. | GCN |
| 2 | 0.9406 | 0.7911 | 0.9850 | 0.9450 |
| 3 | 0.9703 | 0.8208 | 0.9900 | 0.9367 |
| 4 | 0.6445 | 0.5021 | 0.7450 | 0.6875 |
| 5 | 0.7332 | 0.6298 | 0.7960 | 0.7520 |
| 6 | 0.7865 | 0.5283 | 0.8300 | 0.6833 |
| 7 | 0.6899 | 0.5222 | 0.7857 | 0.6471 |

These results demonstrate that our moment-based embeddings consistently outperform the GCN baseline across all levels of mixture complexity. In simpler cases ($k = 2$ and $k = 3$ mixtures), our method achieves perfect or near-perfect clustering. Even when scaled to the maximum tested complexity ($k = 7$ mixtures), our method maintains strong performance, whereas the baseline method's performance degrades substantially.

### F.4 UNDERLYING MODELS

In Figure 8, we present the estimated graphons for the COLLAB and IMDB-BINARY datasets, each corresponding to a cluster identified by our framework in an unsupervised manner. For instance, the estimated graphons display diverse structural patterns for COLLAB: Cluster 8 exhibits a block-like

structure similar to a two-community stochastic block model (SBM) with an imbalanced size ratio, Clusters 2 and 6 display nearly uniform connectivity, suggesting dense or complete graph structures, and Cluster 10 reveals a heavy-tailed pattern indicative of power-law behavior. Although some similarities exist among certain models, the overall variability emphasizes the presence of multiple distinct generative mechanisms within the dataset. This heterogeneity demonstrates the limitations of using a single fixed or random augmentation strategy, as commonly adopted in existing GCL and Mixup methods.

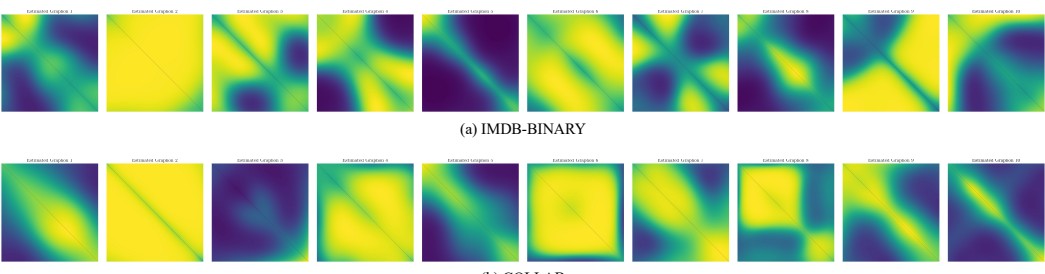

(a) IMDB-BINARY

(b) COLLAB

Figure 8: Cluster-specific estimated graphons in the COLLAB and IMDB-BINARY dataset, revealing diverse structures.

Furthermore, when estimating models within each class for mixup applications, we observe diverse graphons both within and across classes, as illustrated in Figure 9.

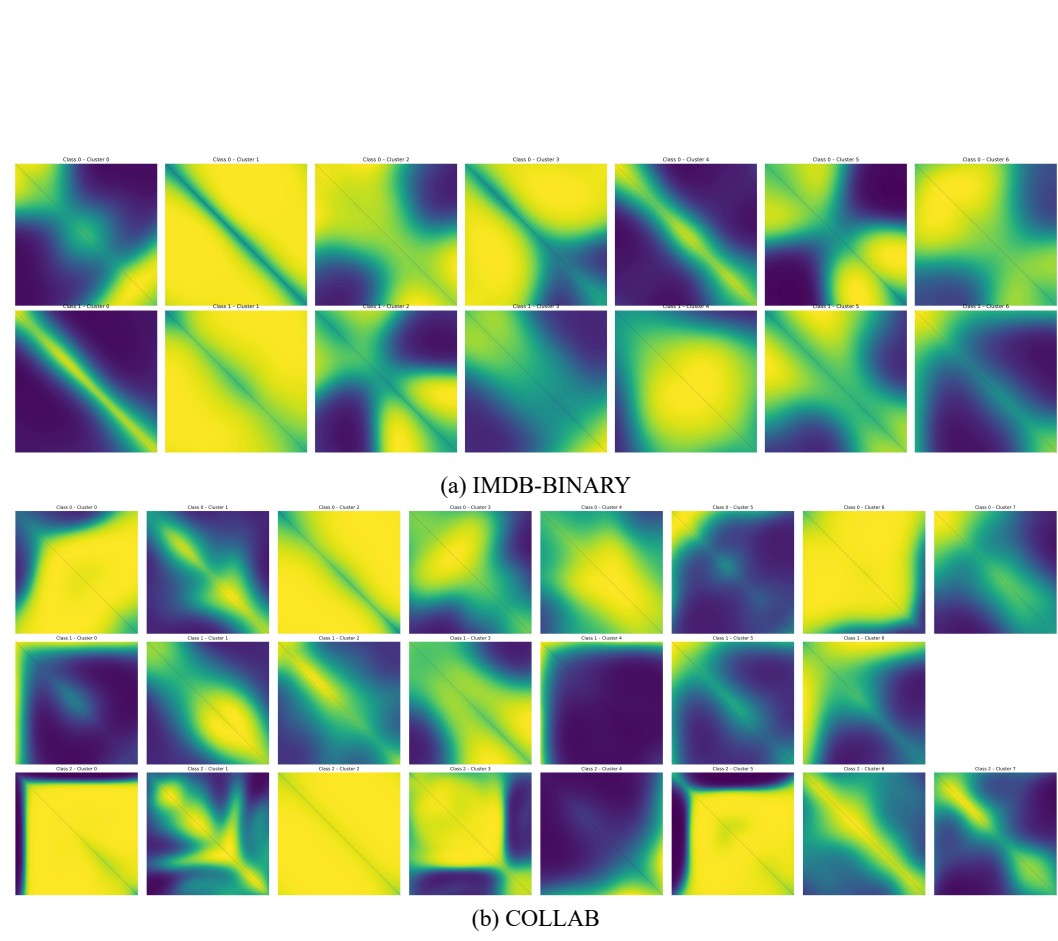

(a) IMDB-BINARY

(b) COLLAB

Figure 9: Cluster-specific estimated graphons in the COLLAB and IMDB-BINARY dataset within each class, revealing diverse structures.

