# OpenReview forum: "From Moments to Models: Graphon Mixture-Aware Mixup and Contrastive Learning"
_ICLR.cc/2026/Conference — Submitted to ICLR 2026_

### Official Review · Reviewer_zrKm · 2025-10-28

**Soundness:** 3
**Presentation:** 3
**Contribution:** 2
**Rating:** 4
**Confidence:** 2

**Summary:**

This paper proposes a unified framework for inferring multiple underlying generative models (i.e., graphon mixtures) from observed graph data and leverages this structure to enhance downstream tasks such as graph mixup augmentation and graph contrastive learning.

**Strengths:**

The work elevates "graph augmentation" from the observation space to the generative space, which is logically self-consistent. Once the graphon estimation is completed, the per-unit training cost is weakly coupled with K, making the computational overhead appear manageable and facilitating easy integration into existing contrastive learning pipelines.

**Weaknesses:**

1. A core idea of this paper is modeling dataset heterogeneity via multiple latent generative factors, which closely resembles the concept of latent factors in disentangled graph representation learning [1, 2]. However, the article lacks comparisons with baselines from this related line of work.

 2. The paper claims to obtain a more disentangled representation but lacks corresponding visualizations or experiments using quantitative disentanglement metrics. For example, visualizations like feature correlation matrices or comparative analyses are missing.

 3. Several choices in the pre-modeling stage (e.g., the selection of K, potential bias in graphon estimation) likely influence the results, yet the paper lacks ablation studies examining these aspects.


[1] Disentangled Graph Contrastive Learning. NeurIPS 2021


[2] Disentangled Graph Convolution Networks. ICML 2019

**Questions:**

See Weaknesses

---

> ### Author Response · Authors · 2025-11-21
> **Rebuttal by Authors**
>
> We appreciate the reviewer’s positive assessment of the conceptual and computational strengths of our work. We thank the reviewer for the time and effort spent evaluating our work. We now provide responses to the raised concerns, and we have updated our submission with additional experiments and explanations highlighted in blue.
>
> **Weaknesses**
>
> **W1)**
>
> We thank the reviewer for raising this point and apologize for any confusion.
>
> While our work indeed models *dataset heterogeneity via multiple latent generative factors*, this objective is different from the goals of the mentioned references.
> Methods such as [1] and [2] aim to disentangle *feature dimensions* within graph embeddings. Their objective is to learn node- or graph-level representations whose latent components are decorrelated or factorized.
>
> In contrast, our goal is not to disentangle the feature space or impose structure on embedding dimensions. Instead, we disentangle the *graph dataset itself* by identifying multiple underlying *generative models* (graphons) that give rise to the observed graphs. Our method recovers mixture components at the level of
> graph *distributions*, not feature factors, and we leverage these recovered generative mechanisms to improve mixup and contrastive learning.
>
> We will revise the paper to more clearly articulate this distinction and avoid any confusion.
>
> Moreover, to address the reviewer’s concern, we additionally include a comparison with DisenGCN. Since DisenGCN is designed for node-level tasks, we extend it with a final aggregation layer to produce graph-level embeddings, and evaluate it in our synthetic clustering experiment alongside other baselines.
>
> As shown in Table 1, DisenGCN performs poorly, failing to separate even simple synthetic datasets, highlighting that feature-level disentanglement is not aligned with the goal of recovering heterogeneous graph generators.
> Our mixture-based method (MBC) remains substantially more effective for this task.
>
> We will include these clarifications and results in the updated version of the manuscript.
>
> **Table 1: Clustering accuracy (%) using different embedding methods.**
>
> | Method | Varying | Fixed |
> | :--- | :---: | :---: |
> | Theory | 81.4 | 82.9 |
> | GCN | 58.6 | 64.7 |
> | GIN | 25.7 | 60.9 |
> | Graph2Vec | 28.6 | 62.0 |
> | DeepWalk | 25.7 | 28.9 |
> | Spectral | 22.9 | 21.7 |
> | **DisenGCN** | **18** | **19.2** |
> | MBC (ours) | **80.0** | **79.3** |
>
> **W2)**
>
> As clarified in our response to the previous concern, our work does *not* aim to obtain a disentangled *feature space* in the sense of decorrelated embedding dimensions. Therefore, visualizations such as feature correlation matrices, commonly used in disentangled representation learning, are not applicable to our setting. Methods referenced in that literature focus on producing factorized feature vectors, which naturally motivates correlation-based diagnostics; in contrast, this is not the objective of our approach.
>
> Instead, our goal is to *disentangle the graph dataset itself* by discovering multiple underlying generative models (graphons) and assigning graphs to these latent components. To visualize this form of disentanglement, we provide (i) the synthetic clustering experiment, where the moment-vector embeddings clearly separate graphs according to their generative sources, and (ii) the graphon visualizations included at the end of the supplementary material, which show the distinct graphons recovered from real-world datasets. These figures directly demonstrate the effectiveness of our method in disentangling heterogeneous graph generators.
>
> **W3)**
>
> We thank the reviewer for this comment.
>
> The influence of pre-modeling choices is indeed important, and we agree that these components should be carefully examined. To address this, our revised appendix now includes a comprehensive set of ablations that directly examine these factors in addition to the existing experiments:
>
> * **Appendix F.1:** Ablation on the number of motifs used in the moment-vector representation.
> * **Appendix F.2.1 and F.2.2:** Ablations on the number of clusters $K$ for both effect on moment embeddings and clustering quality and also on downstream tasks of GMAM and MGCL, showing how downstream performance varies with different cluster counts.
> * **Appendix F.3:** Ablations on the number of mixture components and the impact of varying the number of clusters during graphon mixture estimation.
>
> Additionally, for hyperparameters that are not specific to our method, such as those related to graphon estimation, mixup, or MGCL, we follow the standard settings used in prior work. These details are provided in **Appendix E.2**, ensuring that our comparisons with baselines remain fair and consistent.

---

### Official Review · Reviewer_MyDk · 2025-11-01

**Soundness:** 2
**Presentation:** 3
**Contribution:** 3
**Rating:** 6
**Confidence:** 4

**Summary:**

This work introduces a framework for graph representation learning that models datasets as mixtures of underlying generative processes represented by graphons. The key idea is to represent each latent generative mechanism by a graphon, a continuous function that defines connection probabilities between nodes. To uncover these mechanisms, the authors propose to characterize graphs using motif densities (graph moments), which serve as structural fingerprints. Graphs with similar motif statistics are clustered together, and a distinct graphon is estimated for each cluster. Building on this mixture model, the authors propose two applications: Graphon Mixture-Aware Mixup (GMAM) for semantically consistent data augmentation, and Model-aware Graph Contrastive Learning (MGCL) for reducing false negatives in unsupervised learning. The approach is supported by theoretical analysis and achieves competitive results across several benchmark datasets.

**Strengths:**

**Conceptual novelty** Clearly identifies and formalizes the overlooked “mixture of graphons” problem, which challenges the single-distribution assumption in existing graph learning frameworks.

**Strong theoretical contribution** Introduces a novel, tighter motif concentration bound and provides complete proofs.

**Empirical validation** Demonstrates improvements on both synthetic and real datasets, with extensive ablation and visualization.

**Interpretability** Motif-based clustering yields interpretable “graph fingerprints” and meaningful estimated graphons.

**Clarity and reproducibility** The presentation is very clear, and the appendices provide all implementation details.

**Weaknesses:**

W1: The framework is only evaluated within two settings — Mixup augmentation and contrastive learning. There is no discussion or experiment on extending the mixture-aware framework to other learning paradigms, such as semi-supervised node classification, which essentially corresponds to a subgraph classification task over ego-networks across different hops.

W2: While the proposed methods achieve the best overall results, the performance gains over strong baselines are small, often below 1%, raising concerns about the practical significance of the improvement.

W3：There are minor typos, such as “Equation equation 10” in line 208.

W4: Experiments are confined to small- and medium-scale TUDatasets. It remains unclear how the proposed methods perform on large graphs.

W5: The paper sets the number of mixture components as log of the number of graphs, but the ablation in Appendix shows that performance is quite sensitive to the choice of $K$, suggesting that this prior strategy requires further investigation. In addition, no such ablation is reported for the Mixup setting, where similar sensitivity may arise.

W6: While Appendix presents an ablation on the number of motifs, the paper does not explore how different combinations or types of motifs affect clustering or downstream performance. This leaves open whether the proposed results are robust to motif choice.

**Questions:**

See weakness

---

> ### Author Response · Authors · 2025-11-21
> **Rebuttal by Authors - Part 1**
>
> We thank the reviewer for the positive and thorough evaluation of our work. We are glad that the conceptual novelty of addressing the overlooked mixture-of-graphons setting, the strength of our theoretical contribution, the empirical validation and interpretability, as well as the clarity and reproducibility of the presentation were all appreciated. It is clear from the detailed assessment that the reviewer dedicated significant time to evaluating our manuscript, and we are grateful for this effort.
>
> Below, we address the reviewer’s specific comments and concerns and we have updated our submission with additional experiments and explanations highlighted in blue.
>
> **Weaknesses**
>
> **W1)**
> We thank the reviewer for this comment and the opportunity to clarify our intentions.
>
> First, it is worth emphasizing that our objective is to investigate whether estimating the underlying mixture of graph generative models provides measurable benefits when integrated into existing *graph-level* learning frameworks. This focus aligns with the nature of graphons, which are generative models defined for producing entire graphs. Accordingly, our theoretical results, model formulation, and experiments are all developed in the graph-level setting.
>
> For these reasons, we evaluate our framework within two representative tasks where graphon-based modeling is naturally applicable: (i) Mixup-based graph classification (a supervised graph-level task) and (ii) graph-level contrastive learning (an unsupervised graph-level task). Our intention was to demonstrate that the proposed mixture-aware model can enhance both supervised and unsupervised graph-level pipelines.
>
> Extending graphon mixture estimation to node-level tasks such as semi-supervised node classification is not conceptually straightforward. Node classification typically assumes a *single* observed graph, whereas graphon mixture estimation requires a *collection* of graphs generated from multiple latent models. This fundamental difference makes direct application to node-level settings less applicable within the current formulation.
>
> That said, we agree that the underlying ideas, graphon mixtures, motif-based embeddings, and mixture-aware generative structure, may open the door to broader applications, such as anomaly detection, distribution-shift analysis, or mixture-aware graph resampling. We view our work as an initial step toward principled estimation and use of graphon mixtures, and we are excited about these future directions.
>
> We will revise the manuscript to better discuss these considerations.
>
> **W2)**
> It is important to clarify that the primary goal of our work is not to design new state-of-the-art Mixup or GCL architectures, but rather to investigate whether uncovering the underlying mixture of graphons in heterogeneous graph datasets can yield measurable benefits when incorporated into existing graph-level learning pipelines.
> The fact that our method achieves competitive or state-of-the-art performance supports the validity of the estimated mixtures and demonstrates that model-aware augmentation and contrastive strategies can provide meaningful improvements.
>
> We would also like to emphasize that our approach is conceptually simple, consisting essentially of motif-based embeddings followed by clustering, yet it is able to match or outperform several strong baselines that were *specifically designed* for graph-level mixup or contrastive learning. We view this as evidence that capturing the latent generative structure of graph datasets is indeed beneficial.
>
> Regarding the magnitude of improvement, gains on the order of 0.5--1% are common and often considered meaningful in the Mixup and GCL literature, especially on the widely used benchmark datasets we evaluate. For example, published works such as [1-4] report improvements in this same range when introducing new augmentation or contrastive strategies. Our results align with this trend, while additionally providing a principled generative interpretation.
>
> Finally, we note that we did not aggressively tune our method beyond the standard protocol used by baselines, yet we still observe consistent gains, which further supports that mixture-aware modeling provides genuine benefits rather than overfitting.
>
>
>
> **W3)**
> We thank the reviewer for pointing this out. We apologize for the oversight, and we will correct this typo as well as re-check the manuscript for any similar issues.

---

> > ### Author Response · Authors · 2025-11-21
> > **Rebuttal by Authors - Part 2**
> >
> > **W4)**
> >
> > Our experimental evaluation follows the standard practice in both Mixup and graph contrastive learning, where the TUDatasets serve as the primary benchmark suite. These datasets are also used extensively in prior works such as [1--4], enabling a direct and fair comparison with the existing literature.
> >
> > While some TUDatasets are indeed small, others, such as COLLAB and the REDDIT-{BINARY, MULTI5K, MULTI12K} datasets, are considerably larger, containing thousands of graphs and, collectively, tens of thousands of nodes. These benchmarks are widely regarded as medium-to-large scale within graph-level classification and provide a meaningful testbed for evaluating augmentation and contrastive learning techniques. Importantly, for graph-level tasks, the key requirement is *many* graphs rather than a single extremely large graph.
> >
> > Evaluating very large, single-graph benchmarks (e.g., OGBN datasets) is less aligned with our problem setting, as graphon mixture estimation inherently requires a *collection* of graphs sampled from multiple latent models rather than one massive graph. For this reason, the graph-level datasets used in our experiments are the most appropriate and relevant benchmarks for assessing the proposed framework.
> > Consequently, our selection adheres to established evaluation protocols in the literature, ensuring that our method is rigorously benchmarked against state-of-the-art baselines on a fair and standardized playing field.
> >
> > Please see also our response to Reviewer z9jy’s “Further experiments (4)” comment, where we discuss scalability and the suitability of TUDatasets for graph-level evaluation.
> >
> >
> > **W5)**
> >
> > We thank the reviewer for raising this important point.
> >
> > We agree that performance can vary with the choice of $K$.
> > This is the main reason why we included detailed ablation studies in the appendix.
> > Importantly, notice in Figure 6 that using more than one mixture component consistently outperforms the $K=1$ setting across all datasets. This trend aligns with the central motivation of our work: real-world graph datasets typically arise from multiple underlying generative mechanisms, and enforcing a single-graphon assumption ($K=1$) is overly restrictive. Indeed, the most significant performance drop in our ablations occurs when $K$ is reduced to $1$, confirming that incorporating multiple mixture components is essential. Moreover, in the GCL setting, increasing $K$ also reduces the false negative rate (as illusterated in Figure 5 in Appendix F) by avoiding the grouping of graphs from different latent models into the same cluster, which is a known challenge in contrastive learning.
> >
> > To address the reviewer’s concern for the Mixup setting, we have now added an additional ablation on the number of clusters for GMAM in **Appendix F.2.2**. The results show a similar pattern: although extremely small or extremely large values of $K$ can slightly reduce performance, there is a broad intermediate region where the method is stable. This reinforces the conclusion that the improvements arise from capturing the underlying heterogeneity of the dataset rather than from a specific or finely tuned choice of $K$.
> >
> > **W6)**
> >
> > Our current ablation in Appendix F.1 examines the effect of varying the *number* of motifs used to construct the moment vectors. As discussed there, the performance remains stable once a sufficient number of motifs is included, indicating that the method is not overly sensitive to the dimensionality of the motif representation.
> >
> > Regarding motif *types*, our choice in the main paper follows the standard set of small connected motifs commonly used in prior work [2] on motif-based graph statistics. These motifs provide a balanced mixture of structural patterns (e.g., 2- to 4-node motifs) and are widely adopted because they capture core topological signatures while remaining computationally efficient.
> >
> > Our intention was to keep the framework broadly applicable and lightweight, without introducing additional motif-selection hyperparameters that could complicate the analysis, reduce generality, or substantially increase computational cost.
> >
> >
> > [1] Ling et al. Graph Mixup with Soft Alignments ICML 2023
> >
> > [2] Ramezanpour et al. A Few Moments Please: Scalable Graphon Learning via Moment Matching, NeurIPS 2025
> >
> > [3] Ji et al. Rethinking dimensional rationale in graph contrastive learning from causal perspective, AAAI 2024
> >
> > [4] Suresh et al. Adversarial graph augmentation to improve graph contrastive learning, NeurIPS 2021

---

### Official Review · Reviewer_z9jy · 2025-11-02

**Soundness:** 2
**Presentation:** 3
**Contribution:** 2
**Rating:** 2
**Confidence:** 3

**Summary:**

This submission presents a framework for graph representation learning that models data as a mixture of graphons, using motif density-based clustering to disentangle generative models. It introduces two methods: GMAM (for supervised mixup augmentation) and MGCL (for unsupervised contrastive learning with model-aware sampling).
A theoretical result provides a bound linking the cut distance between graphons and differences in empirical motif densities. Experiments show improved performance over existing mixup and contrastive learning methods.

**Strengths:**

1. The motivation to address graph heterogeneity via graphon mixtures is reasonable and intuitively appealing.
2. The paper is clearly written and well-organized, with good visual aids (e.g., Figure 1) explaining the workflow.
3. Empirical results are generally positive, demonstrating improvements on standard benchmark datasets.

**Weaknesses:**

1. The theoretical component (Theorem 1) is incremental and largely reuses existing concepts from graph theory (e.g., motif density concentration).
The bound provided, although claimed to be tighter, does not appear to yield any substantial new theoretical insight or algorithmic design.
2. Both GMAM and MGCL are relatively straightforward extensions of existing approaches such as G-Mixup, SIGL, and GraphCL.
The modifications mainly add a clustering step based on motif statistics, followed by standard mixup or contrastive loss.
This design is incremental and lacks conceptual depth.
3. The paper only briefly mentions computational complexity in Appendix A.1, without any comparison to baselines or quantitative analysis (e.g., runtime, GPU hours, or scaling with graph size).
Since the proposed methods require motif counting and multiple graphon estimations, the computational overhead is likely significant.
Without this analysis, it is unclear whether the performance gains stem from higher computational cost rather than algorithmic improvement.
4. Further experimental evaluations are needed. 1) No ablation on the number of mixture components or motif types. 2) No sensitivity study to clustering quality or graphon estimation accuracy. 3) The datasets used are relatively small and may not sufficiently stress-test scalability. 4) Missing discussion on training efficiency and memory requirements.

**Questions:**

Could the authors provide an ablation study for GMAM that compares it against a baseline that uses SIGL to estimate a single graphon per class (instead of a mixture)? This would help quantify the specific contribution of the mixture model idea.

---

> ### Author Response · Authors · 2025-11-21
> **Rebuttal by Authors - Part 1**
>
> We thank the reviewer for their assessment of our work. We are glad that the motivation behind modeling graph heterogeneity through graphon mixtures, the clarity and organization of the presentation, and the empirical improvements on benchmark datasets were all appreciated. We also thank the reviewer for the time spent evaluating our manuscript. Below, we provide detailed responses to the raised concerns, and we have updated our submission with additional experiments and explanations highlighted in blue.
>
> **Weaknesses:**
>
> **1) Concern about Theorem 1**
> We thank the reviewer for their feedback. We would like to clarify the novelty and direct algorithmic impact of Theorem 1.
> * Regarding the claim that our work is **"incremental"**, we would like to clarify the novelty of Theorem 1. First, we emphasize that even the baseline $O(k)$ bound (Appendix B.2) is not directly available in the literature; we derived it to establish a theoretical foundation. However, as illustrated in Figure 4 (Appendix B.2), this standard bound is not meaningful for practical analysis. Consequently, we developed a novel two-stage proof technique (Appendix B.1) to derive a substantially tighter $O(\sqrt{k})$ bound, which provides a significant improvement over the derived baseline.
> * Regarding the critique that the bound is "claimed" to be tighter and lacks algorithmic relevance, we wish to clarify that the bound is **rigorously proven**, not claimed. As demonstrated in Figure 4 and Appendix B, our novel proof yields a bound that is objectively tighter than the standard derivation.
>   Furthermore, this theoretical result drives our algorithmic design. Theorem 1 provides the formal guarantee that empirical motif vectors are stable "fingerprints" of their underlying graphons. This stability is precisely what makes our algorithmic pipeline viable. The theorem directly motivates Algorithm 1 (Section 3.1) by justifying clustering in the moment space, which subsequently enables our downstream tasks (GMAM and MGCL). Thus, the bound is not an isolated theoretical exercise; it is the foundational validity condition for our proposed method.
>
> **2) Concern about algorithm desgin**
>
> We thank the reviewer for their comment. We respectfully disagree with the characterization of our work as an "incremental" extension lacking "conceptual depth." Our framework's primary contribution is not the clustering step itself, but its use to solve a major, unaddressed challenge in state-of-the-art graphon estimation.
>
> * **Solving a Core Challenge:** State-of-the-art methods like SIGL (mentioned in **Section 2.1**) and MomentNet are explicitly designed to estimate a *single, unified graphon* for an entire dataset. This is a critical limitation, as real-world data is often a **mixture of populations**. Our paper's core conceptual contribution is a new framework that **disentangles this mixture of generative models** (**Section 3.1**). This is a non-trivial problem, and our "clustering step" is the principled mechanism, theoretically justified by **Theorem 1**, to identify and separate these distinct generative components.
>
> * **Beyond Standard Mixup (GMAM):** Consequently, **GMAM (Section 3.2)** is not a simple extension. Standard G-Mixup assumes one (often incorrect) average graphon per class. Our GMAM performs a more semantically valid mixup by interpolating between the *correct, disentangled generative models* that our framework identifies. This is a fundamentally different and more precise approach, leading to SOTA performance (**Table 2**).
>
> * **Beyond Standard Contrastive Loss (MGCL):** While our contrastive objective follows the general structure used in GCL, we emphasize that this is *standard practice* in the literature: contrastive learning is inherently based on contrasting positives and negatives, and nearly all GCL methods adopt variants of the InfoNCE formulation in graph-level GCL.
>     What is novel in our work is not the outer form of the loss, but the way it is **adapted using the estimated generative models**.
>     To the best of our knowledge, MGCL is the first contrastive framework that incorporates **graphon mixture estimation** to guide the contrastive objective.
>  Specifically, our mixture-aware formulation introduces two contributions that are impossible without estimating the underlying graphons: (i) as shown in **Appendix F** (Figure 5), mixture assignments allow us to **decrease false negatives**; and (ii) the estimated graphons enable a new form of **model-informed augmentation**, where edges are resampled in a way that reflects the underlying generative structure rather than relying purely on random perturbations.
> Thus, although the contrastive loss has the typical structure seen in the GCL literature, its **integration with graphon mixture modeling** represents a substantial conceptual advance and leads to measurable improvements in representation quality.

---

> ### Author Response · Authors · 2025-11-21
> **Rebuttal by Authors - Part 2**
>
> In summary, the "conceptual depth" of our paper lies in moving from a single-model assumption to a more realistic **graphon mixture model**. GMAM and MGCL are just two applications of this mixture-aware paradigm, which we believe opens a new paradigm for future work, such as model-based graph anomaly detection. Furthermore, a key output of our framework is the estimation of meaningful, interpretable graphons for each data cluster (as seen in **Figures 8 and 9**), allowing for novel visualization and statistical analysis of the dataset's underlying structure.
>
>
> **3) Computational Complexity**
> We thank the reviewer for raising this point. It is indeed a valid concern.
> We clarify that Appendix A.1 focuses only on the computational complexity of the graphon-estimation and clustering components, since the computational complexity of the rest of the GMAM and MGCL pipelines (i.e., data augmentation, encoder forward passes, and training) remain unchanged from their respective baselines.
> There is no “baseline” counterpart for the motif+graphon module itself; the appropriate comparison is simply the case where these components are omitted.
>
> To provide a more quantitative picture, we have now conducted a detailed **runtime analysis**, which has been added to Appendix A.2.
> In this new experiment, we measure:
> (i) the time required to compute motif vectors,
> (ii) the time required for clustering and graphon estimation, and
> (iii) the training time of GMAM and MGCL.
> As shown in Figure 3 in Appendix A.2, the additional overhead from motif computation and graphon estimation is consistently smaller than the training time across almost all datasets, with the difference being especially pronounced in datasets containing a larger number of graphs, such as COLLAB, NCI1, and REDDIT-MULTI-5K.
>
> Combined with the performance gains reported in Section 4, these results demonstrate that our improvements are not the result of inflated computational budgets but rather stem from the principled use of graphon mixture estimation. Finally, we note that in our experiments we used SIGL as a high-quality estimator; if runtime were a priority, one could employ faster alternatives such as USVT or SAS.
>
> **4) Further Experiments**
> We thank the reviewer for these comments. Below, we address each point in detail and have added new experiments to Appendix F to support our claims.
>
> ***4.1) Ablation on the number of mixture components or motif types***
>
> We would like to clarify that the ablation on the number of mixture components for **MGCL** was included in **Appendix F.2** in the submission.
> For **GMAM**, a corresponding ablation is provided in our response to the reviewer’s question regarding the use of a single graphon. We have now added an ablation study analyzing the effect of varying the number of clusters in the GMAM setting. Please refer to Appendix F.2.2, as well as our response to your question, for additional details.
>
> We have also now conducted a detailed ablation study using synthetic datasets with an increasing number of underlying mixtures (from 2 to 7).
>
> The complete experiment and results are now presented in **Appendix F.3**. We report performance using the Adjusted Rand Index (ARI) and clustering accuracy. The results are split across two tables for clarity:
> * **Table 9**: Ablation on Number of Mixtures (Varying Size).
> * **Table 10**: Ablation on Number of Mixtures (Fixed Size).
>
> These results (specifically **Tables 9 and 10**) show that our moment-based embeddings consistently **outperform the GCN baseline** across all mixture complexities. For simple cases (2--3 mixtures), our method achieves near-perfect clustering performance (e.g., ARI $\approx 1.000$). Even when scaled to 7 mixtures, our method maintains strong performance, while the baselines degrade substantially.
>
> Furthermore, regarding the ablation study on "motif types", we believe this is already included in our submission.
> Please see Appendix F.1 ('Ablation Study on Number of Motifs'). This section provides exactly this analysis: we start with a single motif and iteratively add more, showing how clustering performance changes.
> The results in the table demonstrate that performance rises sharply and then plateaus after $k=9$ motifs. This justifies our choice of using all connected motifs up to 4 nodes (a 9-dimensional vector) as this set captures the vast majority of the useful information without the computational cost of adding more complex motifs.

---

> > ### Author Response · Authors · 2025-11-21
> > **Rebuttal by Authors - Part 3**
> >
> > ***4.2) Sensitivity study to clustering quality or graphon estimation accuracy***
> >
> > **Clustering Quality**
> > We conducted a dedicated ablation study on the number of clusters, $k$, for our synthetic dataset, which has $7$ ground-truth mixtures, to study the quality of cluster recovery and the tuning of the number of mixture components.
> >
> > * **Location:** This ablation study is now presented in the new **Subsection F.2.1**.
> > * **Results:** We measured cluster quality using the **Adjusted Rand Index (ARI)**, where a higher score indicates a better match to the ground truth. The results are shown in **Table 8**.
> >
> > The results demonstrate that our Moment Embedding's ARI score peaks at $\mathbf{k=6}$ for both datasets (e.g., $0.7166$ ARI for varying size). This result is highly consistent with the ground truth of $7$ mixtures. The peak at $k=6$ rather than $k=7$ is expected, as two of the ground-truth graphons are structurally nearly identical (Gromov-Wasserstein distance of $0.024$, as noted in Section 4.1), and our method successfully groups them together.
> >
> > In sharp contrast, the GCN baseline's quality peaks at $k=5$, and its peak ARI (e.g., $0.5709$ for fixed size) is significantly lower than our method's peak, indicating that it fails to find the correct data structure, unlike our moment-based approach.
> >
> > **Graphon Estimation Accuracy**
> >
> > This is a fair concern, and we thank the reviewer for raising it.
> > Since the ground-truth graphons are unknown for real-world datasets, we cannot directly evaluate graphon estimation accuracy in those settings.
> > To address this, we added a synthetic experiment where the true graphons are available.
> > In this experiment, we compare the estimation quality in two scenarios:
> > (i) the standard setting with a single underlying graphon estimated from observed graphs generated from that single graphon, and
> > (ii) our mixture setting, where graphs are generated from multiple underlying graphons and we estimate a separate graphon for each cluster using our Algorithm 1.
> >
> > As illustrated in Table 1 and Table 2, our method consistently recovers the underlying graphons with low loss across different dataset settings. For instance, in the varying size dataset, we achieve a Gromov-Wasserstein (GW) loss as low as 0.0238 (Cluster 3).
> >
> > When compared to the oracle single-graphon baseline, our mixture model's performance is highly competitive. For example, our estimation for Cluster 2 in the varying size setting yields a GW loss of 0.0272, which is remarkably close to the corresponding single-graphon baseline of 0.0258. This confirms that our cluster-wise estimation strategy effectively disentangles the mixture without significantly compromising estimation quality.
> >
> > **Table 1: Graphon estimation accuracy (GW Loss) for the **Varying Size** dataset mixture compared to the single-graphon baseline.**
> >
> > | Cluster ID | GW Loss (Ours) | GW Loss (Baseline) |
> > | :---: | :---: | :---: |
> > | 0 | 0.0433 | 0.0152 |
> > | 1 | 0.0381 | 0.0258 |
> > | 2 | 0.0272 | 0.0258 |
> > | 3 | 0.0238 | 0.0165 |
> > | 4 | 0.0254 | 0.0212 |
> > | 5 | 0.0280 | 0.0205 |
> > | 6 | 0.0278 | 0.0246 |
> >
> > **Table 2: Graphon estimation accuracy (GW Loss) for the **Fixed Size** dataset mixture compared to the single-graphon baseline.**
> >
> > | Cluster ID | GW Loss (Ours) | GW Loss (Baseline) |
> > | :---: | :---: | :---: |
> > | 0 | 0.0340 | 0.0152 |
> > | 1 | 0.0198 | 0.0212 |
> > | 2 | 0.0294 | 0.0205 |
> > | 3 | 0.0293 | 0.0246 |
> > | 4 | 0.0343 | 0.0168 |
> > | 5 | 0.0301 | 0.0258 |
> > | 6 | 0.0116 | 0.0165 |
> >
> > ***4.3) Concern about size of datasets***
> > Our experimental evaluation follows the standard practice in both Mixup and graph contrastive learning, where the TUDatasets serve as the primary benchmark suite. These datasets are also used extensively in prior works such as [1--4], enabling a direct and fair comparison with the existing literature.
> > While some TUDatasets are indeed small, others, such as COLLAB and the REDDIT-{BINARY, MULTI5K, MULTI12K} datasets, are considerably larger, containing thousands of graphs and, collectively, tens of thousands of nodes. These benchmarks are widely regarded as large-scale within graph-level classification and provide a meaningful testbed for evaluating augmentation and contrastive learning techniques. Importantly, for graph-level tasks, the key requirement is *many* graphs rather than a single extremely large graph.
> > Evaluating very large, single-graph benchmarks (e.g., OGBN datasets) is less aligned with our problem setting, as graphon mixture estimation inherently requires a *collection* of graphs sampled from multiple latent models rather than one massive graph. For this reason, the graph-level datasets used in our experiments are the most appropriate and relevant benchmarks for assessing the proposed framework. Consequently, our selection adheres to established evaluation protocols in the literature, ensuring that our method is rigorously benchmarked against state-of-the-art baselines on a fair and standardized playing field.

---

> > > ### Author Response · Authors · 2025-11-21
> > > **Rebuttal by Authors - Part 4**
> > >
> > > ***4.4) Training efficiency and Memory requirements***
> > >
> > > Please see our response to the "Computational complexity". Regarding memory requirements, the additional memory footprint of our method is minimal. The primary overhead comes from storing the moment embeddings. Since we utilize a fixed set of 9 motifs, this results in a negligible storage cost of just 9 floating-point values per graph (effectively $9 \times \text{Dataset Size}$). This is insignificant compared to the memory required for storing the adjacency matrices used in the subsequent GNN processing steps.
> > >
> > > **Questions:**
> > >
> > > **1) Single Graphon Comparison**
> > >
> > > We thank the reviewer for this helpful suggestion.
> > > We would like to first clarify that this exact setting is already a baseline in our main results.
> > > The baseline the reviewer requests, G-Mixup using SIGL to estimate a single graphon per class, is precisely **'SIGL'** baseline, as presented in Table 2. We define this baseline in **Appendix E.4** as *"which replaces the original graphon estimator in G-Mixup with SIGL"*.
> > >
> > > However, we have included a more general ablation study on the number of clusters for the GMAM setting in **Appendix F.2.2**. In this experiment, we vary the number of clusters used in GMAM to estimate the underlying generative models for each graph class. Importantly, the case of a *single graphon per class*, which corresponds exactly to the baseline suggested by the reviewer, is represented by the setting with **one cluster**.
> > >
> > > We observe that although extremely small or extremely large values of $K$ can slightly reduce performance, there is a broad intermediate region where the method is stable. This reinforces the conclusion that the improvements arise from capturing the underlying heterogeneity of the dataset rather than from a specific or finely tuned choice of $K$.
> > >
> > > This ablation directly quantifies the contribution of the mixture-model idea: GMAM with mixture-aware graphon estimation consistently outperforms the single-graphon-per-class baseline (i.e., SIGL with one cluster), thereby validating the importance of modeling the heterogeneity present within each class.
> > >
> > >
> > > [1] Ling et al. Graph Mixup with Soft Alignments ICML 2023
> > >
> > > [2] Ramezanpour et al. A Few Moments Please: Scalable Graphon Learning via Moment Matching, NeurIPS 2025
> > >
> > > [3] Ji et al. Rethinking dimensional rationale in graph contrastive learning from causal perspective, AAAI 2024
> > >
> > > [4] Suresh et al. Adversarial graph augmentation to improve graph contrastive learning, NeurIPS 2021

---

### Author Response · Authors · 2025-12-04
**Discussion Summary**

For added convenience, we summarize the main concerns raised by the reviewers along with our corresponding answers and manuscript updates. We thank all reviewers for their constructive feedback, and we thank the area chair for considering our work.

**Reviewer z9jy**

1. *The reviewer questioned the novelty of Theorem 1 and the conceptual depth of our method*.
   - We *clarified the novelty* of our theoretical bound, explaining that it is derived through a *new two-stage proof technique* (Appendix B) that, to our knowledge, does not appear in prior literature.
   - We explained how Theorem 1 supports the core algorithmic design, specifically how *motif–moment stability justifies clustering* in Algorithm 1.
   - We *expanded the conceptual comparison* against both single-graphon estimators and disentangled-feature approaches (related concerns from Reviewer zrKm).

2. *The reviewer requested quantitative evidence regarding computational overhead and scalability*.
   - We added a *detailed runtime analysis* in *Appendix A.2*, measuring motif extraction, clustering, and graphon estimation time compared to full GMAM/MGCL training time.
   - As shown in *Appendix A.2*, the overhead from motif + graphon estimation is *consistently smaller* than the end-to-end training cost across nearly all datasets.

3. *The reviewer asked for additional ablations and sensitivity analyses*.
   - We added several new ablations:
     - *Appendix F.1:* Ablation on the number of motifs (included in the original submission).
     - *Appendix F.2.2:* Ablation on the number of clusters $k$ for downstream tasks (GMAM and MGCL).
     - *Appendix F.2.1:* Ablation on the estimated number of clusters in the synthetic $k$-component dataset, evaluating recovery quality.
     - *Appendix F.3:* Ablation on the number of *true* mixture components, demonstrating robustness as mixture complexity increases.

4. *The reviewer asked whether our mixture-wise graphon estimates are accurate*.
   - We added a new experiment comparing our *cluster-wise graphon estimates* with an *oracle single-graphon baseline* on synthetic data where the true graphons are known.
   - Results show *low GW loss*, with mixture-wise estimates performing *competitively with the oracle*.

5. *The reviewer asked about dataset size and scalability*.
   - We emphasized the value in using well-known datasets for evaluating a novel Mixup or GCL method.
   - We clarified that graphon mixture estimation requires *multiple graphs*, making TU datasets the standard benchmark suite.
   - Large TU datasets (e.g., *COLLAB*, *REDDIT-MULTI-12K*) contain thousands of graphs and tens of thousands of nodes.

**Reviewer MyDk**

1. *The reviewer noted that our evaluation focuses only on Mixup and GCL*.
   - We clarified why graphon models naturally apply to *graph-level* learning tasks and explained why extending to node-level settings is nontrivial within the graphon mixture framework.

2. *The reviewer commented that performance gains appear small*.
   - We contextualized these gains within the *Mixup and GCL literature*, where improvements of similar magnitude are typical and meaningful.
   - We emphasized that our method is intentionally simple yet still competitive with specialized baselines.

3. *The reviewer expressed concerns about the dataset size*.
   - Addressed jointly with Reviewer z9jy: TU datasets are standard for graph-level tasks and sufficiently large for evaluating mixture-aware models and well-known for GCL and Mixup methods.

4. *The reviewer asked about sensitivity to the number of mixture components $k$*.
   - We added a new *GMAM ablation* in *Appendix F.2.2*.

6. *The reviewer asked about motif choices*.
   - Addressed via *Appendix F.1*, which analyzes the effect of the number of motifs.
   - We also expanded the discussion of *standard connected motifs up to 4 nodes*, explaining why they provide a good balance of expressiveness and tractability.

**Reviewer zrKm**

1. *The reviewer raised concerns about the relation to disentangled representation learning and corresponding visualizations*.
   - We clarified the conceptual distinction: Our method targets *dataset-level generative disentanglement* (graphon mixtures), not *feature-space disentanglement*.
   - We added a *DisenGCN comparison* in Table 3, showing that feature-disentangling approaches do not perform well for graphon mixture recovery.

2. *The reviewer questioned whether modeling choices significantly affect performance*.
   - This is addressed by our expanded ablations in *Appendix F.1-F.3*, showing robustness to number of motifs, number of clusters, and number of true mixture components.

We thank all reviewers again for their time and constructive suggestions.
We believe that the extensive clarifications and newly added experiments thoroughly address the concerns raised.

---

### Meta-Review · Area_Chair_vYEf · 2026-01-13

**Summary:**

Reviewer z9jy holds that the theoretical contribution is incremental, the model design lacks conceptual depth,  and the paper omits both a quantitative analysis of computational complexity and necessary ablation studies and discussions.  Reviewer MyDk holds that there is a lack of discussion or experiment on extending the mixture-aware framework to other learning paradigms,  the performance gains achieved are marginal, and the paper omits necessary ablation studies under different mixup settings.  Reviewer zrKm holds that there is a lack of comparisons with baselines from the cited line of research, as well as quantitative evaluations using established disentanglement metrics.

**Reviewer Concerns:**

After rebuttal, the computational complexity has been thoroughly analyzed, several ablation studies have been added, and minor typos have been corrected. The theoretical contributions remain limited, and the model design remains conceptually superficial.

**Reviewer Scores:**

Reviewer z9jy may maintain the original score, as he/she holds that the theoretical contribution is minor, and the model design is trivial.  Reviewer MyDk  may maintain the original score, as he/she concerns the experiments on large graphs, and the performance gains are marginal. Reviewer zrKm  may maintain the original score, as he/she holds that the proposed model is closely resembles the concept of latent factors in disentangled graph representation learning while the comparisons with the baselines from this related line of work is missing.

---

### Decision · Program_Chairs · 2026-01-26

Reject